# High-frequency stimulation-induced peptide release synchronizes arcuate kisspeptin neurons and excites GnRH neurons

Jian Qiu[1]*[†], Casey C Nestor[1†], Chunguang Zhang[1], Stephanie L Padilla[2], Richard D Palmiter[2], Martin J Kelly[1,3]*[‡], Oline K Rønnekleiv[1,3]*[‡]

[1]Department of Physiology and Pharmacology, Oregon Health and Science University, Portland, United States; [2]Department of Biochemistry, Howard Hughes Medical Institute, University of Washington, Seattle, United States; [3]Division of Neuroscience, Oregon National Primate Research Center, Oregon Health and Science University, Beaverton, United States

*For correspondence: qiuj@ohsu.edu (JQ); kellym@ohsu.edu (MJK); ronnekle@ohsu.edu (OKR)

[†]These authors contributed equally to this work
[‡]These authors also contributed equally to this work

**Abstract** Kisspeptin (Kiss1) and neurokinin B (NKB) neurocircuits are essential for pubertal development and fertility. Kisspeptin neurons in the hypothalamic arcuate nucleus (Kiss1[ARH]) co-express Kiss1, NKB, dynorphin and glutamate and are postulated to provide an episodic, excitatory drive to gonadotropin-releasing hormone 1 (GnRH) neurons, the synaptic mechanisms of which are unknown. We characterized the cellular basis for synchronized Kiss1[ARH] neuronal activity using optogenetics, whole-cell electrophysiology, molecular pharmacology and single cell RT-PCR in mice. High-frequency photostimulation of Kiss1[ARH] neurons evoked local release of excitatory (NKB) and inhibitory (dynorphin) neuropeptides, which were found to synchronize the Kiss1[ARH] neuronal firing. The light-evoked synchronous activity caused robust excitation of GnRH neurons by a synaptic mechanism that also involved glutamatergic input to preoptic Kiss1 neurons from Kiss1[ARH] neurons. We propose that Kiss1[ARH] neurons play a dual role of driving episodic secretion of GnRH through the differential release of peptide and amino acid neurotransmitters to coordinate reproductive function.

## Introduction

Pulsatile secretion of gonadotropin-releasing hormone 1 (GnRH), and consequently pulsatile release of pituitary luteinizing hormone (LH), is required for normal reproductive function, and there is increasing evidence that kisspeptin (Kiss1) neurons in the hypothalamic arcuate nucleus (ARH) are a component of the "GnRH pulse generator" (*Navarro et al., 2009*; *Navarro et al., 2011a*; *Lehman et al., 2010a*; *Okamura et al., 2013*). The "GnRH pulse generator" consists of bursts of multiunit activity (MUA) in the arcuate nucleus that are timed with pulses of LH secretion, and this phenomenon has been demonstrated in the monkey (*Knobil, 1989*), goat (*Wakabayashi et al., 2010*; *Okamura et al., 2013*), and rat (*Kimura et al., 1991*; *Kinsey-Jones et al., 2008*). In mammals, two major populations of kisspeptin-synthesizing neurons exist - one in the anterior preoptic area and the other in the ARH (*Lehman et al., 2010b*). In rodents, the anterior preoptic cell group is a periventricular continuum within the anteroventral periventricular nucleus (AVPV) and the periventricular preoptic nucleus (PeN) and is responsible for the GnRH/LH surge (*Clarkson and Herbison, 2006*).

**eLife digest** Puberty and fertility are necessary for survival of the species. An evolutionarily ancient region of the brain called the hypothalamus regulates these processes. The hypothalamus releases a chemical messenger called gonadotropin-releasing hormone (or GnRH for short), which is then transported from the brain to the pituitary gland. GnRH activates the pituitary gland, which in turn releases reproductive hormones that control ovulation in females and sperm production in males.

For this process to work correctly in females, the hypothalamus must release GnRH in appropriately timed pulses and produce one massive release, or "surge", of GnRH to trigger ovulation. Two populations of neurons within the hypothalamus produce a peptide molecule called Kisspeptin and drive the activity and subsequent release of GnRH. One population resides in an area called the arcuate nucleus and the other in the preoptic nucleus. Recent findings suggest that the arcuate nucleus is the "pulse generator" responsible for triggering the rhythmic release of GnRH by the hypothalamus, whereas the preoptic nucleus induces the surge of GnRH. However, how these brain regions do this remains unclear.

Using a technique called optogenetics, Qiu, Nestor et al. explored whether kisspeptin-producing neurons in the arcuate nucleus are able to communicate with each other to drive pulses of GnRH release. The idea was to selectively activate a subset of kisspeptin neurons in mice and determine whether this would activate the remaining neurons at the same time. Qiu, Nestor et al. introduced a light-sensitive protein into the kisspeptin-producing neurons on one side of the arcuate nucleus, and then used light to activate those neurons. As predicted, this caused kisspeptin neurons throughout the arcuate nucleus to coordinate their activity.

In addition to their namesake peptide, kisspeptin-producing neurons also make the neurotransmitter glutamate, the excitatory peptide neurokinin B, and the inhibitory peptide dynorphin. Light-induced stimulation of the arcuate nucleus caused its kisspeptin neurons to also release neurokinin B and dynorphin, which synchronized the firing of the kisspeptin neurons. The hypothalamus then translates this coordinated activity into pulses of GnRH release. The light-induced stimulation also triggered the release of glutamate, which caused kisspeptin neurons within the preoptic nucleus to fire in bursts. This in turn robustly excited the GnRH neurons, giving rise to the surge of GnRH.

These findings show that peptide and classical neurotransmitters collaborate to control GnRH neuron activity and, consequently, fertility. The results obtained by Qiu, Nestor et al. can be used to further explore kisspeptin-GnRH neuronal circuits, and to obtain insights into the role of neuronal peptide signaling in healthy as well as diseased states.

Both the AVPV/PeN and ARH Kiss1 neurons (Kiss1$^{AVPV/PeN}$; Kiss1$^{ARH}$) co-express other peptide and amino acid neurotransmitters, and the Kiss1$^{ARH}$ neurons co-synthesize and potentially co-release neurokinin B (NKB) and dynorphin (Dyn) in most mammals (*Goodman et al., 2007*; *Navarro et al., 2011b*; *Bartzen-Sprauer et al., 2014*), forming the basis for the "KNDy neuron" terminology (*Lehman et al., 2010a*). Importantly, Kiss1, NKB and their respective receptors are absolutely essential for pubertal development and the control of reproduction (*Seminara et al., 2003*; *d'Anglemont de Tassigny et al., 2007*; *Topaloglu et al., 2009*). Morphological studies have provided evidence that Kiss1$^{ARH}$ neurons can communicate with each other (*Lehman et al., 2010a*; *Rance et al., 2010*; *Navarro et al., 2011a*; *Navarro et al., 2011b*), and NKB/tachykinin receptor 3 (Tacr3) and Dyn/κ-opioid receptor (KOR) signaling have been proposed to play a critical role in the generation of episodic GnRH/LH pulses via direct and synchronized action on GnRH nerve terminals in the median eminence (*Navarro et al., 2009*; *Ohkura et al., 2009*; *Wakabayashi et al., 2010*). Anatomical tract-tracing studies have shown that Kiss1$^{ARH}$ neurons are bilaterally interconnected within the ARH and project to GnRH terminals in the median eminence and to Kiss1$^{AVPV/PeN}$ neurons (*Rance et al., 2010*; *Yip et al., 2015*). Therefore, we hypothesized that Kiss1$^{ARH}$ neurons could coordinate (synchronize) their activity to drive GnRH secretion via these contralateral projections. Recent findings also indicate that Kiss1$^{ARH}$ neurons co-express the vesicular glutamate

transporter 2 (vGluT2) and release the neurotransmitter glutamate (*Cravo et al., 2011*; *Nestor et al., 2016*), which could also provide an excitatory drive to hypothalamic neurons (*Van den Pol et al., 1990*).

Currently, we explored the synaptic mechanisms by which Kiss1$^{ARH}$ neurons synchronize their burst firing activity and how their unified activity was conveyed to the rostral Kiss 1$^{AVPV/PeN}$ neurons and hence excitation of GnRH neurons. We used a combination of whole-cell patch recording, optogenetics, molecular pharmacology and single cell RT-PCR to elucidate the underlying synaptic mechanisms at multiple sites. We discovered that high-frequency photoactivation of Kiss1$^{ARH}$ neurons induced a NKB-mediated slow excitatory postsynaptic potential (EPSP), which was modulated by co-released dynorphin, and caused synchronization of Kiss1$^{ARH}$ neurons. Interestingly, activation of Kiss1$^{ARH}$ neurons co-released glutamate that excited the rostral Kiss1$^{AVPV/PeN}$ neurons. The combination of synchronization and unified excitation of Kiss1$^{AVPV/PeN}$ neurons resulted in robust excitation of GnRH neurons.

## Results

### Frequency and duration-dependent activation of a slow EPSP in Kiss1$^{ARH}$ neurons

To investigate the firing properties of Kiss1 neurons necessary for peptide release, we did whole-cell recordings of photostimulated ARH and AVPV/PeN Kiss1 neurons expressing ChR2:YFP in slices obtained from OVX females (*Figures 1A–C, 2A–D*). These neurons were confirmed to express Kiss1 mRNA based on scRT-PCR (*Figure 2D*). High fidelity responses to light (470 nm) were measured as evoked inward currents in voltage clamp ($V_{hold}$ = -60 mV) or depolarizations in current clamp (*Figures 1* and *2*). A 10-s photostimulation at 1 or 5 Hz had no lasting effect on Kiss1$^{ARH}$ neuron activity; however, with 10 Hz stimulation there was a prominent depolarization of ~10 mV that peaked about a minute later, and it was accompanied by a train of action potentials during that time (*Figure 2E*). We also varied the duration of photostimulation at 20 Hz, which revealed that 1 s was ineffective but with 5 or 10 s of stimulation there was again a prominent depolarization accompanied by a barrage of action potentials that continued for about a minute after stimulation (*Figure 2F*). The results from varying frequency and/or duration of photostimulation revealed that a maximum depolarization of 22 ± 1.4 mV was obtained with a 10 s stimulation at 20 Hz (*Figure 2G,H*). We refer to this depolarization as a slow excitatory postsynaptic potential (EPSP). To determine whether the slow EPSP is a unique property of Kiss1$^{ARH}$ neurons, we also tested the AVPV/PeN population of Kiss1 neurons with 10-s stimulation at 20 Hz (*Figure 2I*); these neurons had a transient hyperpolarization after stimulation and a return to continuous firing ~30 s after the stimulation. The depolarization of Kiss1$^{ARH}$ neurons was also distinct from POMC and AgRP/NPY neurons in the ARH, even though they are probably all derived from a common precursor (*Sanz et al., 2015*). The relatively quiescent POMC neurons fired action potentials during the 10-s, 20-Hz stimulation and exhibited a prolonged hyperpolarization afterwards (*Figure 2J*); whereas the spontaneously active AgRP neurons exhibited a brief pause in firing in conjunction with a slight hyperpolarization (*Figure 2K*). Thus, Kiss1$^{ARH}$ neurons are unique among the neurons tested in the generation of the slow EPSP. Given that the expression of Kiss1, dynorphin, NKB and Tacr3 are all down-regulated in kisspeptin neurons by E2-treatment (*Smith et al., 2005*; *Gottsch et al., 2009*; *Navarro et al., 2011a*; *Ruiz-Pino et al., 2012*), we asked the question whether the slow EPSP would also be reduced by E2 or even vary during the ovulatory cycle. The slow EPSP was induced in slices obtained from diestrous or proestrous animals or OVX females treated with either oil vehicle or E2. The slow EPSP was the most robust in OVX females (13.8 ± 1.6 mV, n= 17), which was to be expected if this current was peptidergic in nature—i.e., correlates with the up-regulation of NKB (*Figure 2L*). Moreover, in vivo E2 treatment resulted in a significant reduction of the slow EPSP amplitude (4.8 ± 1.1 mV, n = 9, p<0.001). Importantly, the slow EPSP was also of greater magnitude in Kiss1$^{ARH}$neurons from diestrous animals (9.2 ± 1.1 mV, n=14) and significantly reduced in Kiss1$^{ARH}$ neurons from proestrous animals (3.3 ± 0.8 mV, n=15, p<0.01) (*Figure 2L*).

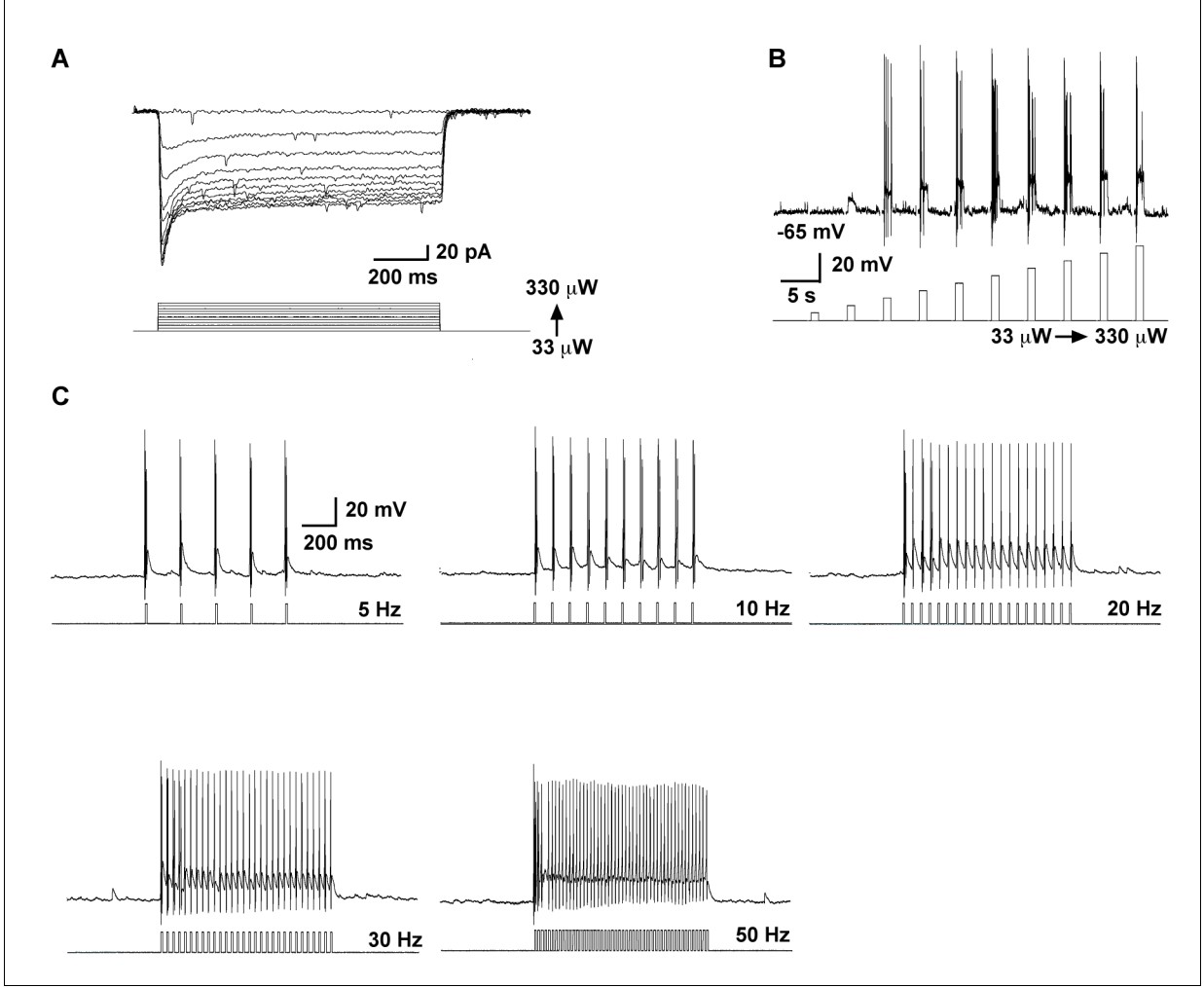

**Figure 1.** High fidelity response to light (470 nm) stimulation of ARH Kiss1 ChR2-expressing neurons. (**A**–**B**) Examples of evoked inward currents in voltage-clamp (**A**) and depolarization in current-clamp (**B**) with varied light intensity (33–330 µW). (**C**) Trains of light pulses (0.9 mW, 10 ms pulse-width) at various frequencies and the resulting action potentials in the same neuron from **A** and **B**.

## The slow EPSP is dependent on the direct synaptic input from Kiss1 neurons

To investigate the properties of the slow EPSP, we used a ratio method in which the same neuron was stimulated (10 s, 20 Hz) and then tested again 10 min later after experimental manipulation (*Figure 3A*). Under control conditions the R2/R1 ratio was 0.65 ± 0.04 (n = 12) (*Figure 3A,F*). The slow EPSP was abolished by perfusing CSF containing low $Ca^{2+}$ (0.5 mM) and high $Mg^{2+}$ (10 mM) or tetrodotoxin (TTX, 0.5 µM) (*Figure 3B,D,F*). We could rescue TTX blockade of the light-induced response with the addition of $K^+$ channel blockers 4-aminopyridine (4-AP, 0.5 mM) and tetraethylammonium (TEA, 7.5 mM) to facilitate ChR2- mediated depolarization of nerve terminals and neurotransmitter release (*Cousin and Robinson, 2000*; *Petreanu et al., 2009*), arguing that the response is dependent on direct pre-synaptic input from neighboring Kiss1 neurons (*Figure 3C,F*). Although the majority of Kiss1[ARH] neurons are glutamatergic (*Cravo et al., 2011*), release glutamate locally in ARH (*Nestor et al., 2016*) and the fast EPSP was blocked by CNQX (10 µM) and AP5 (50 µM), the slow EPSP was unaffected by these ionotropic glutamate receptor antagonists (*Figure 3E,F*). Action potentials were abrogated by dialyzing Kiss1[ARH] neurons with the $Na^+$ channel blocker QX314 (0.5 mM in the internal solution) (*Isaac and Wheal, 1993*). The expression of the slow EPSP was reduced but not blocked by QX314 (*Figure 4A,E*). The reduction in the amplitude of the slow EPSP

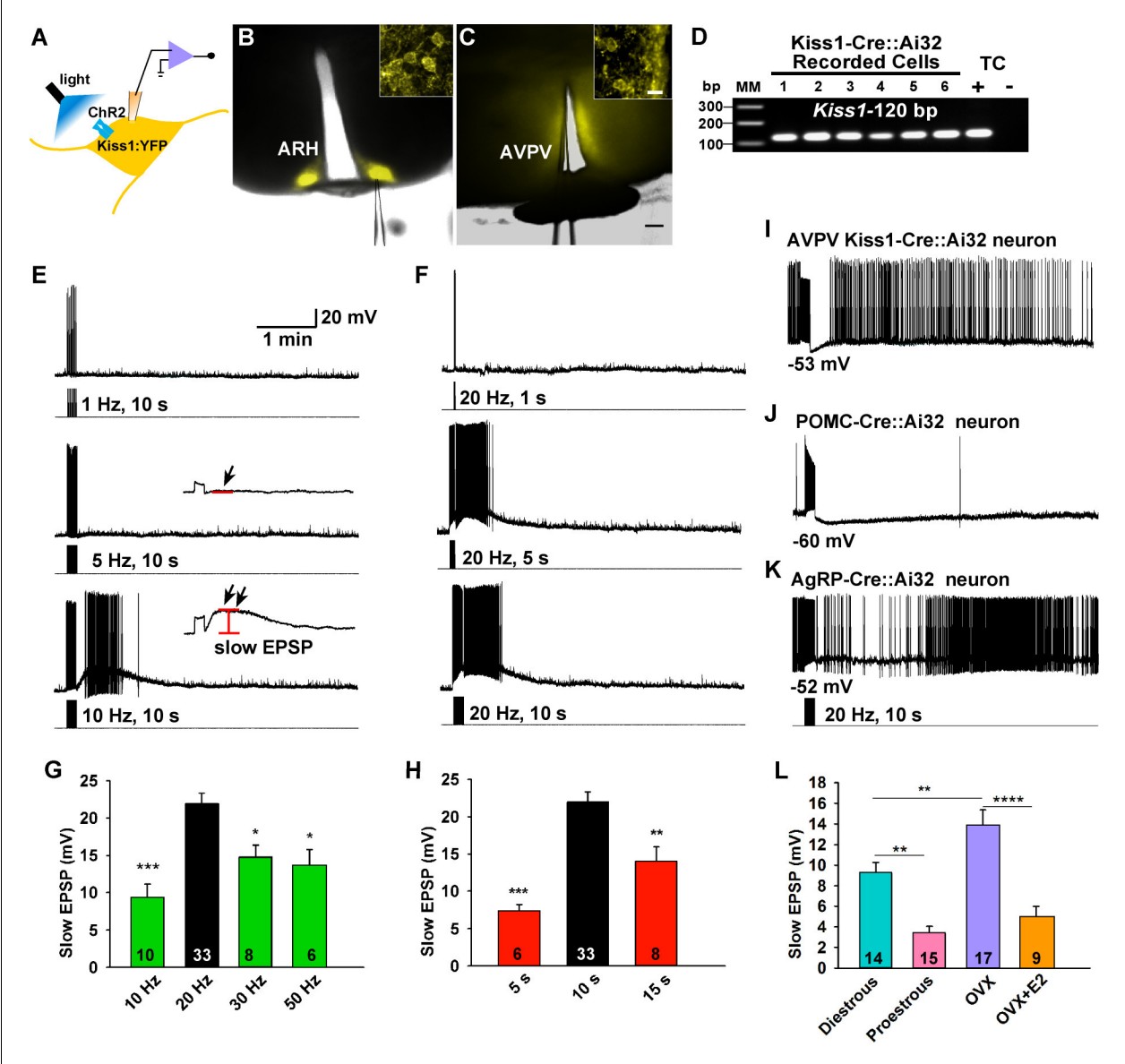

**Figure 2.** Slow excitatory postsynaptic potential (slow EPSP) is frequency and duration dependent. (A) Illustration of experimental approach. (B–C), Overlay of epifluorescence (YFP) and differential interference contrast (DIC) images during whole-cell recording (pipette tip can be seen) from Kiss1 neurons in ARH (B) and AVPV (C) brain slices from a Kiss1-Cre::Ai32 mouse (scale bar = 200 μm). (B,C) Insets show high power magnification of labeled cells in each area (scale bar = 10 μm). ARH, hypothalamic arcuate nucleus; AVPV, anteroventral periventricular nucleus. (D) Confirmation of *Kiss1* expression in YFP-expressing neurons from Kiss1-Cre::Ai32. Representative gel illustrating single-cell RT-PCR identification of *Kiss1* mRNA following whole-cell recording in ChR2-YFP neurons. RNA extracted from the medial basal hypothalamic tissue control (TC) was included as positive (+, with RT) and negative (-, without RT) controls. MM, molecular marker. (E) Slow EPSP induced by a 10-s photostimulation (light intensity 0.9 mW and pulse duration, 10 ms) with varied frequencies in the same neuron. The insets show the measurement of slow EPSP after low-pass filtering. 1–5 Hz photostimulation did not induce any post-stimulus depolarization (**arrow, middle trace**); but ≥10 Hz stimulation generated a significant post-stimulus depolarization (**double arrow, lower trace**). (F), Examples of synaptic responses induced by a train of stimuli (0.9 mW, 10 ms pulse-width) delivered at 20 Hz with varied duration in the same Kiss1$^{ARH}$ neuron. (G–H), Bar graphs summarizing slow EPSP responses induced by a train of stimulation at 10, 20, 30 and 50 Hz with duration of 10 s (G) and by a 20 Hz-stimulation delivered at 5, 10 and 15 s (H), current-clamped to -70 mV. The slow EPSP was larger when induced by 10-s and 20 Hz photostimulation (one-way ANOVA, effect of treatment, $F_{(3, 53)}$ = 9.912, p<0.0001 (G); $F_{(2, 44)}$ = 12.69, p<0.0001 (H); Newman-Keuls's Multiple-comparison test, ***, **, * indicates p<0.005, 0.01 and 0.05, respectively. (I–K), Slow EPSP is unique to arcuate Kiss1 neurons. (I) Photostimulation induced auto-inhibition in AVPV Kiss1 neuron, (J) ARH POMC neuron and (K) AgRP neuron from Kiss1-Cre::Ai32, POMC-Cre::Ai32 and AgRP-Cre::Ai32 mice, respectively. (L) Light evoked slow EPSP in Kiss1$^{ARH}$ neurons was reduced by E2 treatment and varied during the ovulatory cycle. Bar graphs summarizing slow EPSP responses induced by 20 Hz, 10 s photostimulation in slices obtained from diestrous or proestrous females, or OVX females treated with either oil vehicle or E2 that had received injection of AAV-DIO-ChR2:YFP into ARH. Slow EPSP was larger in low estrogen

*Figure 2 continued on next page*

*Figure 2 continued*

states (one-way ANOVA, effect of treatment, F $_{(3, 51)}$ = 15.43, p<0.0001; Newman-Keuls's Multiple-comparison test. ****, ** indicates p<0.001 and 0.01, respectively.

was probably due to QX314 suppression of calcium currents in the Kiss1 neurons (*Hu et al., 2002*). Moreover, postsynaptic electrical stimulation (20 Hz, 10 s) of individual neurons, even in the presence of 4-AP and TEA, did not mimic the light-induced EPSP (*Figure 4B,C,E*), suggesting that the response was not dependent on the generation of action potentials in the cell soma.

## The slow EPSP is mediated by the G protein-coupled receptor Tacr3

We have measured *Tacr1* and *Tacr3*, but not *Tacr2*, mRNA expression in Kiss1$^{ARH}$ neurons using single cell RT-PCR (*Navarro et al., 2015*), and the Tacr3 agonist senktide depolarizes Kiss1$^{ARH}$ neurons (*Navarro et al., 2011b*). Firstly, in order to inhibit G protein activation we dialyzed Kiss1 neurons with the GDP analog GDP-β-S (Guanosine 5'-[β-thio]diphosphate trilithium, 2 mM in the internal solution) and completely abrogated the response (*Figure 4D and E*), demonstrating that the slow

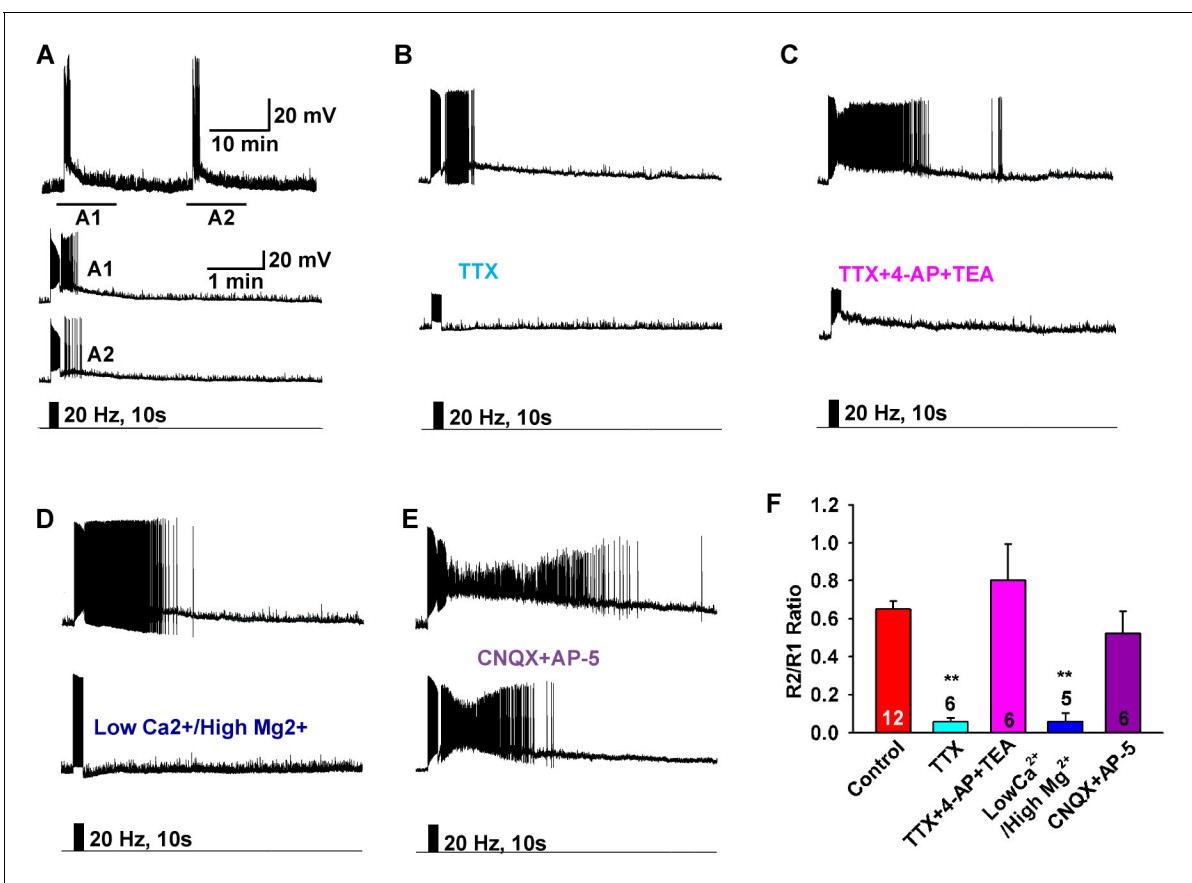

**Figure 3.** Slow EPSP is dependent on direct synaptic input from neighboring Kiss1 neurons. (**A**) To reduce the variability between Kiss1 neurons from different animals, a response ratio (R2/R1) protocol was used in which the magnitude of second light response (A2) was compared to first response (A1). Representative traces showing that the R2/R1 ratio of two photostimuli is 0.65 in Kiss1$^{ARH}$ neurons. (**B–E**), Representative traces showing that the slow EPSP was abolished by perfusing TTX (**B**), and the TTX blockade was rescued by the addition of the K$^+$ channel blockers 4-AP and TEA, suggesting a direct synaptic input from other Kiss1 neurons (**C**); the slow EPSP was abolished by perfusing low Ca$^{2+}$/high Mg$^{2+}$ (**D**); and the slow EPSP was unaffected by the ionotropic glutamate receptor antagonists CNQX and AP5 (**E**). (**F**) Bar graphs summarizing the effects of drugs on the direct synaptic input (EPSP) to Kiss1 neurons. Comparisons between different treatments were performed using a one-way ANOVA analysis (F $_{(4, 30)}$ = 12.18, p<0.0001) with the Newman-Keuls's *post hoc* test. ** indicates p<0.01 vs. control.

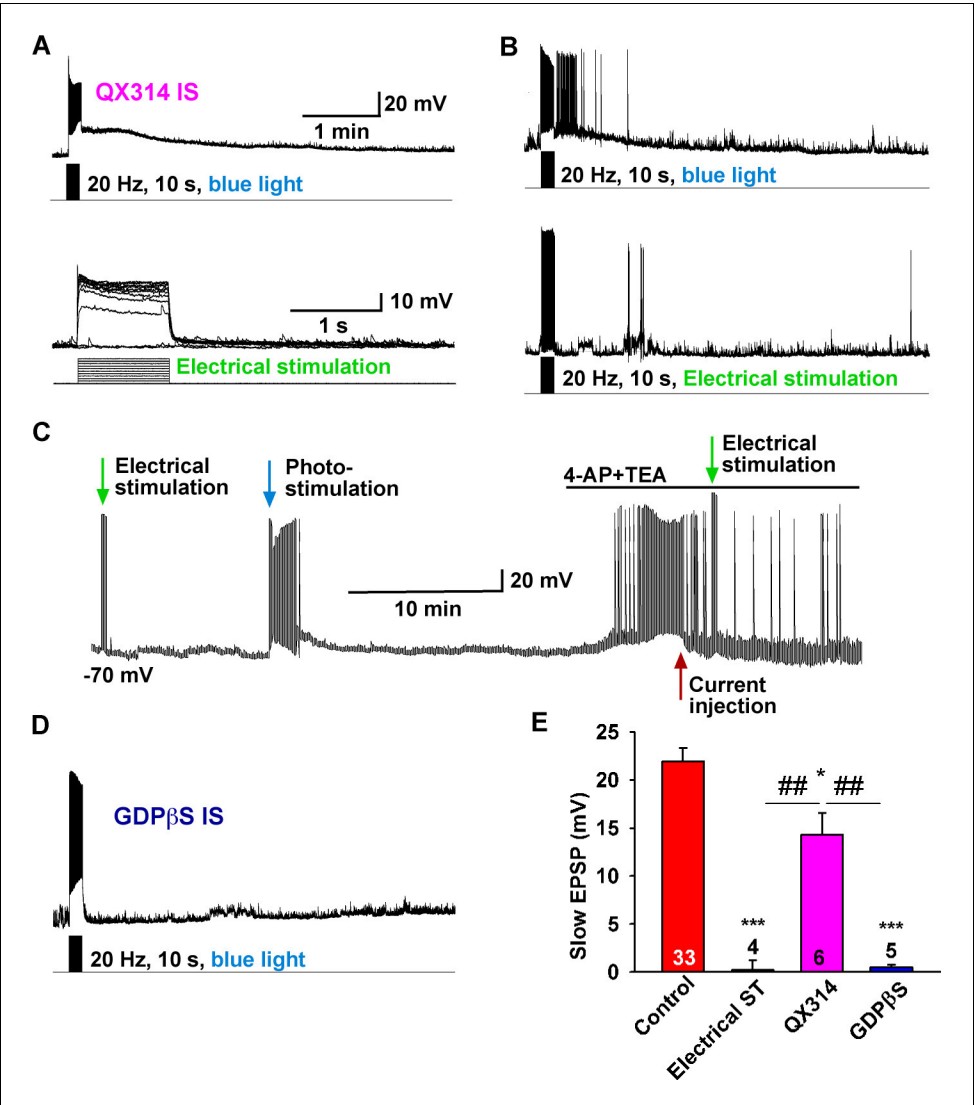

**Figure 4.** The slow EPSP is dependent upon G-protein activation. (**A**) Intracellular dialysis with low concentrations of QX-314 (0.5 mM) in the internal solution (IS) blocked Na$^+$ channels, but the slow EPSP induced by light stimulation was not blocked. (**B**) Slow EPSP induced by photostimulation (**upper**) was not mimicked by electrical stimulation (**lower**) at 20 Hz for 10 s in the same cell. (**C**) Electrical stimulation in the presence of 4-AP and TEA did not mimic the effects of photostimulation to induce a slow EPSP in Kiss1$^{ARH}$ neurons; a hyperpolarizing bias (current injection) was used to repolarize membrane potential to resting membrane potential when the 4-AP/TEA response reached the maximal depolarization. (**D**) Intracellular dialysis with GDPβS (2 mM) blocked the slow EPSP, demonstrating that the slow EPSP was mediated by G protein-coupled receptors. (**E**) Bar graphs summarizing the effects of Na$^+$ channel and G-protein blockers and electrical stimulation on the slow EPSP in Kiss1 neurons. Comparisons between different treatments were performed using a one-way ANOVA analysis (F$_{(3, 44)}$ = 22.39, p<0.0001) with the Newman-Keuls's *post hoc* test. *** and * indicates p<0.005 and p<0.05 vs. control, respectively; ## indicates p<0.01, QX314 group vs electrical stimulation and GDPβS groups.

EPSP was mediated by a G protein-coupled receptor like Tacr3. Secondly, we blocked the slow EPSP with a Tacr3 antagonist SB222,200 (3 µM) or with a cocktail of antagonists against all 3 tachykinin receptors (**Figure 5A,B,D**). The slow EPSP was also occluded by pretreatment with Tacr3 agonist senktide that depolarized Kiss1$^{ARH}$ neurons and pharmacologically prevented any effects of the photostimulated release of NKB (**Figure 5C–D**). Finally, To see if dual photostimulation induced desensitization, we perfused senktide repeatedly (2x), and the averaged ratio of the responses (0.65 ± 0.01,

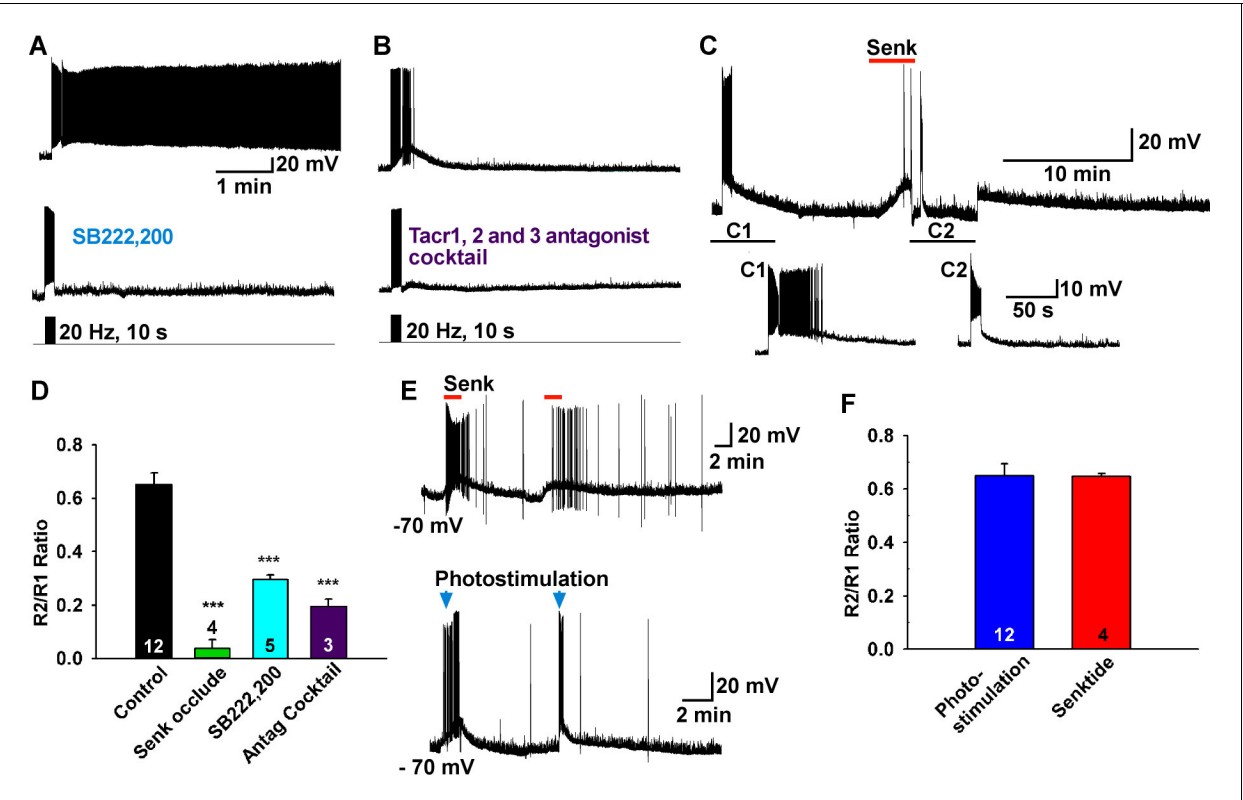

**Figure 5.** Tacr3 agonist mediates the slow EPSP. (**A–C**) The slow EPSP was abrogated by an NKB receptor (Tacr3) antagonist SB222,200 (**A**) and the NKB receptor antagonist cocktail SDZ-NKT 343 (1 µM), GR94,800 (1 µM) and SB222,200 (3 µM) (Tacr1, 2 and 3 antagonists, respectively) (**B**). Furthermore, Tacr3 agonist senktide pretreatment completely occluded the slow EPSP (**C**), demonstrating that the slow EPSP was mediated by the Tacr3. A hyperpolarizing bias was used to repolarize the membrane potential to −70 mV when the senktide response reached the maximal depolarization (C2). (**D**) Bar graphs summarizing the effects of Tacr3 agonist and antagonists on the slow EPSP. Comparisons between different treatments were performed using a one-way ANOVA analysis ($F_{(3, 20)}$ = 31.6, p<0.0001) with the Newman-Keuls's *post hoc* test. *** indicates p<0.005 vs. control. (**E–F**) Depolarization induced by senktide (250 nM) perfused on two separate occasions in a Kiss1[ARH] neuron. The R2/R1 ratio was 0.65 (**E**, **upper trace**), which was not different from the ratio from photostimulation (**E**, **lower trace**). (**F**) Bar graphs summarizing R2/R1 ratios (Un-paired t-test, $t_{(14)}$= 0.0408, p = 0.9680).

n = 4) was similar to that obtained by photostimulation, suggesting that desensitization was caused by internalization and downregulation of Tacr3 following binding of NKB (*Steinhoff et al., 2014*) (*Figure 5E,F*).

## The slow EPSP is modulated by endogenous dynorphin

Since dynorphin is co-localized to over 90% of Kiss1[ARH] neurons (*Goodman et al., 2007*; *Navarro et al., 2009*), and κ-opioid receptor mRNA is expressed in a subset of Kiss1[ARH] neurons (*Navarro et al., 2009*), we hypothesized that dynorphin would be co-released with high-frequency photostimulation and modulate NKB release from Kiss1 terminals. Indeed, blockade of κ-opioid receptors by the selective antagonist nor-BNI (1 µM) potentiated the slow EPSP (*Figure 6A,C*). Moreover, the κ-opioid receptor agonist U69,593 (1 µM) abrogated the slow EPSP through its pre-synaptic actions, but Kiss1[ARH] neurons (n = 4) still responded directly to senktide (*Figure 6B*). Finally, U69,593 (1 µM) reduced the frequency but not the amplitude of miniature EPSCs (mEPSCs); U69,593 did not alter the holding current (or input resistance) of Kiss1[ARH] neurons (*Figure 6D–F*). Therefore, co-released Dyn is acting presynaptically to inhibit the peptidergic (NKB) and glutamatergic excitatory input to Kiss1[ARH] neurons.

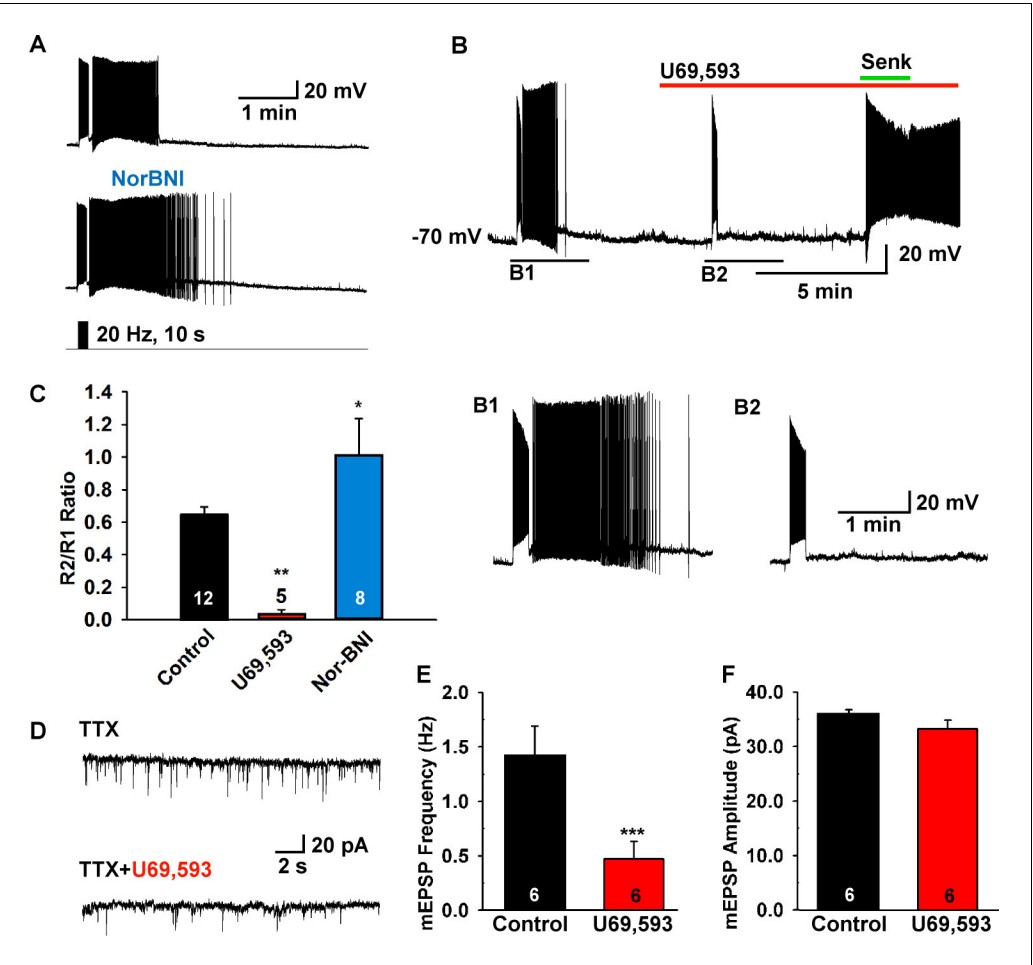

**Figure 6.** The κ-opioid receptor agonist blocks the slow EPSP. (A–B) The κ-opioid receptor antagonist nor-BNI (1 μM) potentiated the slow EPSP (A) and the agonist U69,593 (1 μM) attenuated the slow EPSP (B2 versus B1). However, the Tacr3 agonist senktide is able to excite Kiss1[ARH] neurons in the presence of U69,593, indicative that the kappa-opioid effects are pre-synaptic to inhibit NKB release (B). (C) Bar graphs summarizing the effects of κ-opioid receptor agonist and antagonist on the slow EPSP. Comparisons between different treatments were performed using a one-way ANOVA analysis (F $_{(2, 22)}$ = 9.784, p = 0.0009) with the Newman-Keuls's *post hoc* test. ** and * indicates p<0.01 and p<0.05 *vs.* control, respectively. (D–F) Effects of U69,593 on mEPSCs in Kiss1[ARH] neurons. Representative traces showing mEPSCs recorded at a holding potential of −60 mV under control conditions (TTX), and after TTX plus 1 μM U69,593. Summary data showing the effects of U69,593 on the frequency (E) and the amplitude (F) of mEPSCs in Kiss1[ARH] neurons (n = 6). Paired t-test, $t_{(5)}$ = 8.104, ***p<0.001 (E) and Paired t-test, $t_{(5)}$=1.580, p = 0.1750 (F) compared to the control.

## Kiss1[ARH] neurons directly communicate with contralateral Kiss1[ARH] neurons

To study the ChR2-expressing neuronal input to non-ChR2-expressing Kiss1 neurons, an AAV-DIO-ChR2:mCherry virus was unilaterally injected into one side of the ARH (*Figure 7A*). As expected, these injections labeled mCherry-positive (Kiss1) cells and fibers ipsilaterally, but only mCherry-positive fibers in the contralateral ARH (*Figure 7A*). In addition to the ChR2:mCherry labeling ipsilaterally, control AAV-DIO-YFP was injected into the contralateral ARH in order to identify Kiss1 neurons for whole cell recordings (*Figure 7B–D*). With low-frequency photostimulation of the ChR2 fibers in the contralateral ARH (*Figure 7E–G*), we detected a glutamatergic fast EPSC in voltage clamp (22.9 ± 8.3 pA, n=6) (*Figure 7E*, **upper trace**) and with high-frequency stimulation a peptidergic slow EPSP in current clamp (19.2 ± 4.1 mV, n = 6) (*Figure 7E*, **lower trace**). There was no ChR2-mediated

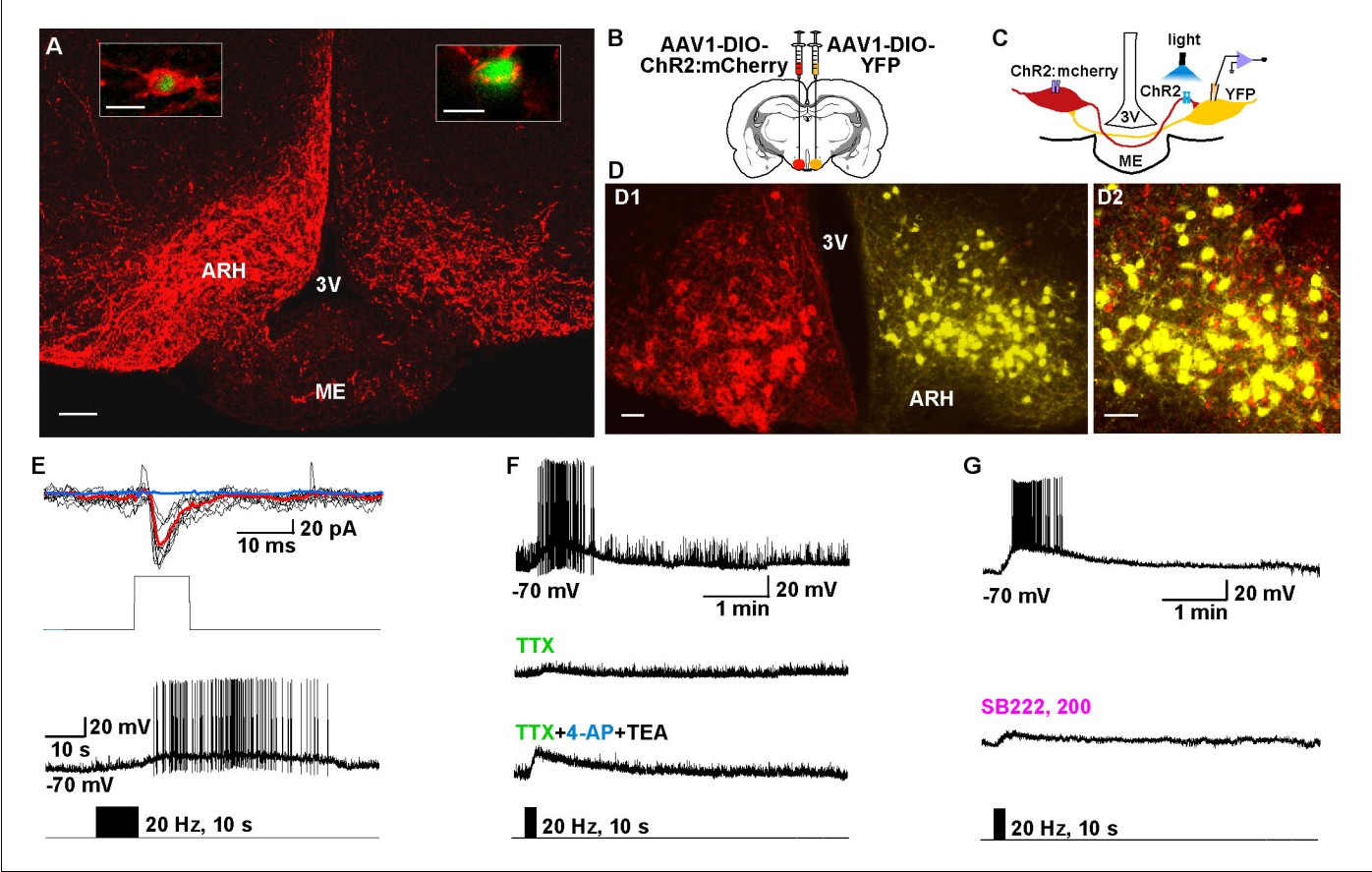

**Figure 7.** Kiss1[ARH] neurons send contralateral projections to Kiss1[ARH] neurons inducing circuit excitation. (**A**) Confocal image of a coronal section through ARH from a Kiss1-Cre mouse following a unilateral ARH injection of AAV-DIO-ChR2:mCherry. Kiss1 neurons expressing ChR2-mCherry were observed on the injected left side, whereas only fibers were visible in the internal zone of the median eminence (ME) and in the contralateral ARH. (Scale bar = 100 μm). Insets on the ipsilateral and contralateral sides illustrate high power images of dual labeling of Kiss1-GFP and mCherry. On the ipsilateral side, a representative Kiss1-GFP neuron show co-expression of GFP and mCherry. (Scale bar = 10 μm). On the contralateral side, mCherry-positive fibers formed close contacts with a representative Kiss1-GFP neuron. (Scale bar = 10 μm). (**B**), Schematic of coronal sections through the ARH illustrating ChR2-mCherry and control YFP injections, in which one side of the ARH was injected with AAV-DIO-ChR2:mCherry and the contralateral side with AAV-DIO-YFP. (**C**) Illustration of the experimental approach. The whole-cell recorded neurons were YFP- expressing neurons. (**D**) Higher power images of a coronal section through the ARH from a Kiss1-Cre female showing the Kiss1 neurons expressing ChR2-mCherry (**left side**, red) (**D1**), versus AAV-DIO-YFP (**right side**, yellow) and overlay of epifluorescence (YFP and mCherry) images in YFP positive neuron area showing mCherry-expressing terminals (**D2**). Only mCherry expressing fibers were observed in the contralateral ARH (**D2**). Scale bars = 20 μm. ARH, hypothalamic arcuate nucleus; 3V, third ventricle. (**E**) Representative trace showing that photostimulation induced not only a fast glutamate response (**upper traces**) with a single 10 ms pulse, but also a slow EPSP (**lower trace**) with 20 Hz for 10 s photostimulation in the same cell. Red trace indicates averaged sweeps. Blue trace illustrates that the AMPA- and NMDA-mediated currents (EPSCs) were blocked with CNQX (10 μM) and D-AP5 (50 μM). (**F**) Representative traces showing that the slow EPSP was abolished by perfusing TTX, and the TTX blockade was rescued by the addition of non-selective K[+] channel blockers 4-AP and TEA suggesting a direct input from other Kiss1 ChR2-expressing neurons. All Kiss1 neurons (n = 7) were rescued by the combination of K[+] channel blockers. (**G**) Blue light stimulation induced a slow EPSP in a contralateral Kiss1[ARH] neuron (**upper trace**), which was antagonized by SB222,200 (3 μM) (**lower trace**).

inward current and/or depolarization with photostimulation. The slow EPSP was blocked by TTX and rescued by TTX plus 4-AP and TEA, demonstrating direct communication between Kiss1[ARH] neurons on both sides (**Figure 7F**). Similar to the recordings in ipsilateral ChR2-expressing Kiss1[ARH] neurons, the Tacr3 antagonist SB222,200 (3 μM) blocked the optogenetic response (reduced R2/R1 by 75% from 0.64 ± 0.06, n = 4 to 0.16 ± 0.04, n = 5; Un-paired t-test, t(7) = 7.193, ****p<0.001) (**Figure 7G**). Moreover, with dual patch recording we were able to simultaneously depolarize and excite the contralateral Kiss1[ARH] neurons (n = 2 pair of neurons) following photostimulation (**Figure 8A–C**). To ensure that the response was due to activation of the axonal terminal field of

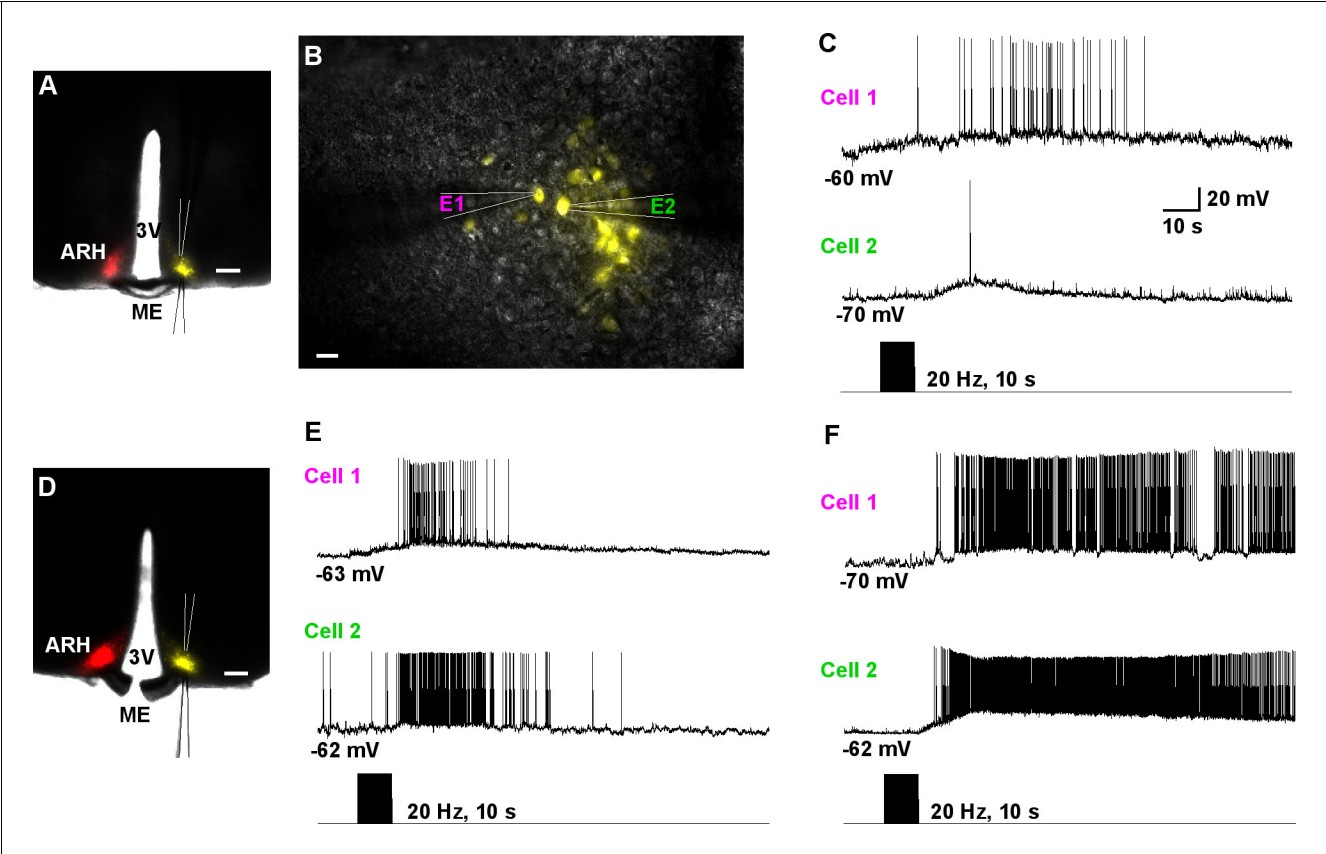

**Figure 8.** Dual patch recording reveal that high-frequency auto-excitation of ipsilateral Kiss1[ARH] neurons recruit contralateral Kiss1[ARH] neurons. (**A**) Image of a coronal section through the ARH from Kiss1-Cre mouse that received dual injection of AAV-DIO-ChR2:mCherry (**left side**, red) and AAV-DIO-YFP (**right side**, yellow). Scale bar = 200 μm. (**B**) DIC and fluorescent image (YFP) of dual patch recording from Kiss1[ARH] neurons in the contralateral ARH; whole cell recordings were made with electrode 1 (**E1**) and electrode 2 (**E2**) in the current-clamp mode. Scale bar = 20 μm. (**C**) High-frequency photostimulation induced a simultaneous increase in firing and a slow EPSP in a pair of Kiss1[ARH] neurons (Cell 1 and Cell 2) recorded in the contralateral ARH. (**D**), Image of a coronal section through the ARH from Kiss1-Cre mouse that received dual injection of AAV-DIO-ChR2:mCherry (**left side**, red) and AAV-DIO-YFP (**right side**, yellow). In this slice, the ME was cut to disconnect the two sides. Scale bar = 200 μm. (**E** and **F**) Examples of synchronized activity obtained using dual patch recording from two pairs of Kiss1[ARH] neurons in the contralateral ARH (**D**). ARH, hypothalamic arcuate nucleus; ME, median eminence; 3V, third ventricle.

ChR2-projecting Kiss1[ARH] neurons, we acutely cut the median eminence to isolate the contralateral Kiss1[ARH] neurons, which did not express ChR2, and found that high-frequency photostimulation of the ChR2-expressing terminal field simultaneously depolarized and increased the firing of Kiss1[ARH] neurons (n = 3 pair of neurons) (*Figure 8D–F*). Thus, high-frequency autoexcitation of Kiss1[ARH] neurons ipsilaterally is able to recruit Kiss1[ARH] neurons bilaterally to induce synchronization of this critical neural network.

## The Tacr3 agonist senktide excites GnRH neurons indirectly and is mimicked by optogenetic stimulation of Kiss1[ARH] neurons

In a number of animal models including mouse, GnRH neurons are primarily located in the POA, and the mechanism by which Kiss1[ARH] neurons regulate GnRH neurons is not well understood (*Navarro et al., 2015*). Currently, we tested whether pharmacological activation of Tacr3 within Kiss1-GnRH circuits would directly and/or indirectly excite GnRH neurons (*Figure 9A,C*). The Tacr3 agonist senktide robustly excited GnRH neurons in the horizontal slice which included both ARH kisspeptin and the preoptic GnRH neurons (*Figure 9A,B*), but not in the coronal slice in which the ARH kisspeptin neurons were separated from the preoptic (rostral) GnRH neurons (*Figure 9C,D*). As expected, kisspeptin excited GnRH neurons irrespective of slice orientation (*Figure 9B,*

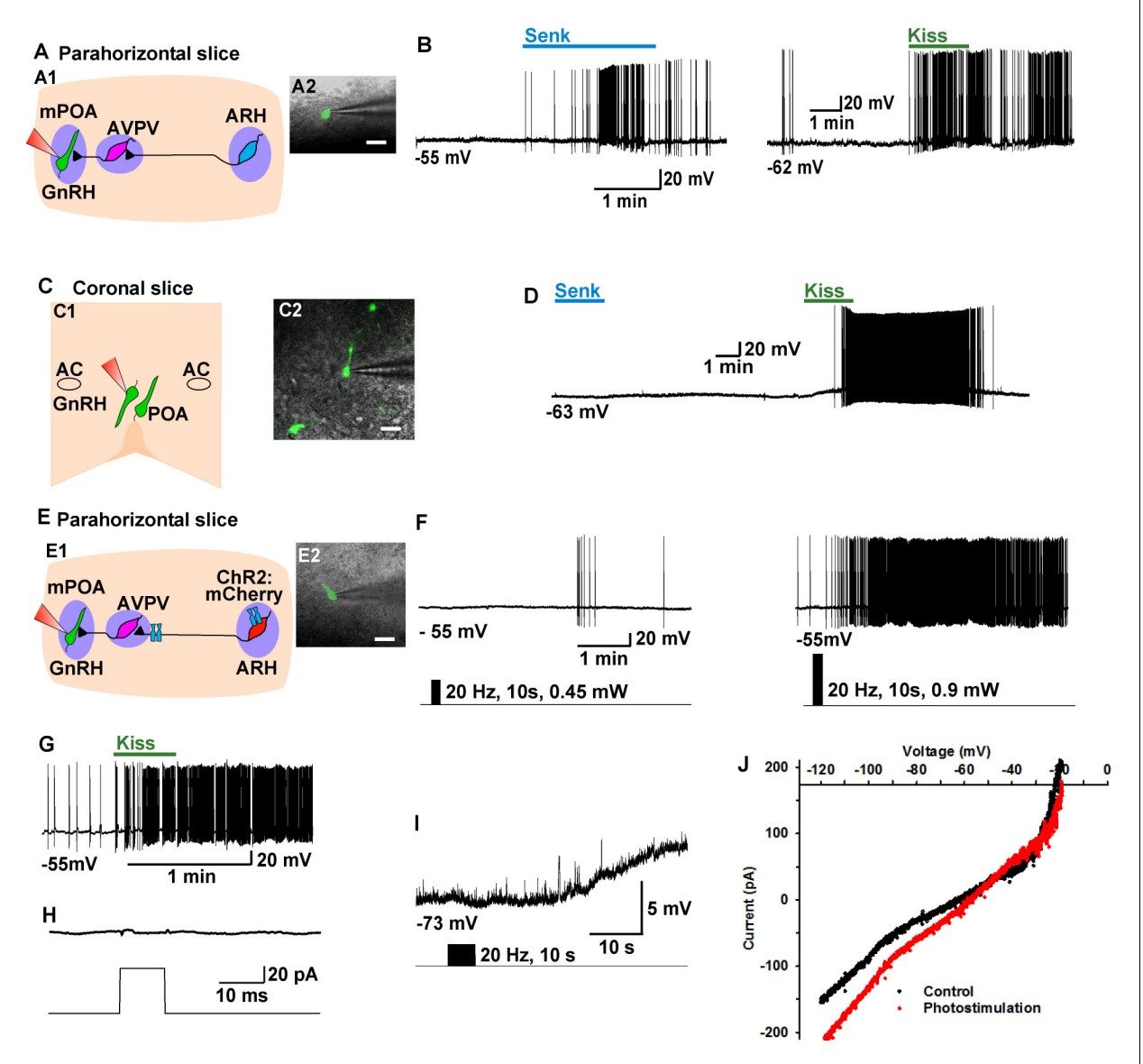

**Figure 9.** Tacr3 agonist senktide excites GnRH neurons indirectly through ARH Kiss1 neurons, and a slow EPSP in GnRH neurons is evoked with photoactivation of ChR2 Kiss1[ARH] neurons. (**A**) Schematic drawing of the Kiss1[ARH], Kiss1[AVPV/PeN] and GnRH neuronal circuit (**A1**) and a high power image (**A2**) of a parahorizontal slice from a GnRH-EGFP mouse illustrating whole-cell recording of GnRH neuron. Scale bar = 20 μM. (**B**) Representative traces showing that perfusing the slice with senktide (1 μM) induced depolarization and increased firing in a GnRH neuron (**left**), which was mimicked by perfusion with kisspeptin (10 nM, **right**). (**C**) Schematic drawing (**C1**) and highpower image (**C2**) of a coronal slice through the POA from a GnRH-EGFP mouse illustrating whole-cell recording of GnRH neuron. (**D**) Representative trace showing that perfusing the slice with senktide (1 μM) did not depolarize or increase firing of a GnRH neuron, but kisspeptin could excite the same POA GnRH neuron. **E,** Schematic drawing of the Kiss1[ARH], Kiss1[AVPV/PeN] and GnRH neuronal circuit (**E1**) and a high power image (**E2**) of a parahorizontal slice from a bilateral arcuate AAV-DIO-ChR2:mCherry injected Kiss1-Cre::GnRH-EGFP mouse illustrating whole-cell recording of GnRH neuron. Scale bar, 20 μM (**E2**). (**F–G**) High-frequency photoactivation of ChR2-labeled ARH neurons with low (0.45 mW, **left panel**) and higher (0.9 mW, **right panel**) intensities evoked slow EPSPs and increased firing in the same GnRH neuron (**F**) which was mimicked by perfusion with kisspeptin (10 nM) (**G**) but low-frequency photostimulation (0.5 Hz) did not evoke a fast EPSC in this GnRH neuron (**H**). (**I–J**) High-frequency photoactivation of ChR2 evoked a slow EPSP (**I**) in another GnRH neuron, and the I-V relationship before and during the peak response from the same cell indicated that the reversal potential of the nonselective cation current was ~ -40 mV (**J**). AC, anterior commissure; ARH, hypothalamic arcuate nucleus; AVPV, anteroventral periventricular nucleus; mPOA, medial preoptic area.

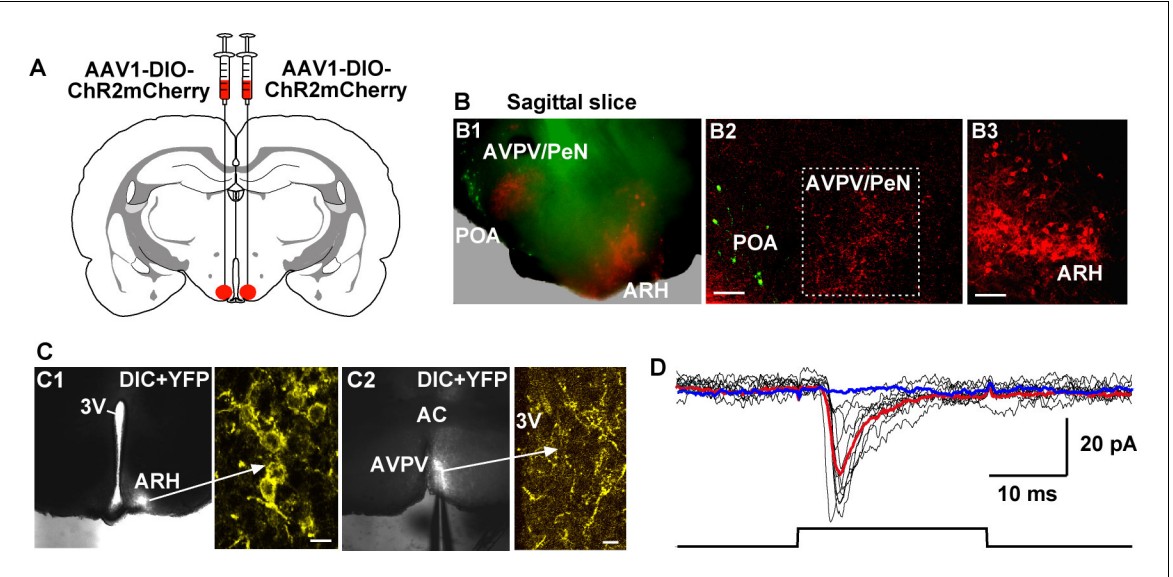

**Figure 10.** Photoactivation of Kiss1[ARH] fibers excites Kiss1[AVPV/PeN] neurons via glutamate release. (**A**) Schematic drawing of a coronal section showing the bilateral viral injections in the ARH with AAV-DIO-ChR2:mCherry. (**B**) Sagittal brain slice (**B1**) and confocal images from the slice (**B2, 3**) of mouse bilaterally injected with a ChR2-mCherry-expressing viral vector into the ARH of Kiss1-Cre::GnRH-EGFP female mouse. Projections from Kiss1[ARH] neurons (**B3**) to the AVPV/PeN are illustrated (**B2**). Green neurons in the POA illustrate the location of GnRH neurons in the sagittal brain slice. Scale bars, 100 µm (**B2, B3**). (**C**) Overlay of epifluorescence (YFP) and differential interference contrast (DIC) images of a coronal arcuate slice showing injection site (**C1**) in a Kiss1-Cre:ChR2:YFP mouse (arrow points to high power image of labeled neurons in the ARH) and fiber-projections to the AVPV (**C2**; arrow points to labeled fibers in the AVPV area that are being photostimulated). Scale bars = 10 µm. (**D**) Example of light-evoked fast EPSCs in a Kiss1[AVPV/PeN] neuron (mean response: 33.2 ± 9.7 pA, n = 5). Red trace indicates averaged sweeps. Blue trace illustrates that the AMPA- and NMDA-mediated currents (EPSCs) were blocked with CNQX (10 µM) and D-AP5 (50 µM). Following recording, cells were harvested for scRT-PCR determination of Kiss1 mRNA expression. All recorded AVPV/PeN cells (n = 5) expressed Kiss1 mRNA (data not shown). ARH, hypothalamic arcuate nucleus; AVPV/PeN, anteroventral periventricular and periventricular preoptic nuclei; POA, preoptic area.

**D**). Therefore, NKB/Tacr3 signaling is upstream from kisspeptin/GPR54 signaling, and activation of kisspeptin/GPR54 signaling via Tacr3 stimulation results in excitation of GnRH neurons.

As proof of principle in a horizontal slice that preserved the connections between the Kiss1[ARH] neurons, Kiss1[AVPV/PeN] neurons and the preoptic area containing GnRH neurons, we found that high-frequency (20 Hz) photostimulation of Kiss1[ARH] neurons depolarized and excited GnRH neurons (5.5 ± 1.4 mV, n = 5) via activation of a non-selective cation channel with double rectifying properties typical of TRPC channels (*Zhang et al., 2008*) (*Figure 9E-F, I-J*), and this excitation was mimicked by bath application of 10 nM kisspeptin (*Figure 9G*). Importantly, there were no fast EPSCs detected in GnRH neurons with photostimulation of Kiss1[ARH] ChR2- expressing neurons (*Figure 9H*).

## Kiss1[ARH] neurons excite AVPV/PeN Kiss1 neurons via glutamate

Given that Tacr3 is essentially not expressed in GnRH or in Kiss1[AVPV/PeN] neurons, and Kiss1[AVPV/PeN] neurons do not respond to kisspeptin (*Navarro et al., 2015*; *Ducret et al., 2010*), the question became how does activation of Kiss1[ARH] neurons excite GnRH neurons? A recent study has shown that mouse Kiss1[ARH] neurons project to Kiss1[AVPV/PeN] neurons (*Yip et al., 2015*). Indeed, we recorded light-evoked EPSCs in Kiss1[AVPV/PeN] neurons (33.2 ± 9.7 pA, n = 5) following photostimulation of fibers from ChR2-mCherry or from ChR2-YFP expressing Kiss1[ARH] neurons (*Figure 10A–D*). The fast EPSC was blocked with CNQX (10 µM) and D-AP5 (50 µM), documenting a glutamatergic input from Kiss1[ARH] neurons to Kiss1[AVPV/PeN] neurons (*Figure 10D*). Furthermore, we bilaterally injected ChR2:mCherry-expressing viral vector into the AVPV/PeN of Kiss1-Cre::GnRH-EGFP female mice. The anterograde transport revealed close contacts with GnRH cell bodies in the mPOA (*Figure 11A, B1–3*). Viral-labeled Kiss1[AVPV/PeN] neurons also revealed projections caudally to the ARH (*Figure 11B4*). High-frequency photostimulation of the Kiss1[AVPV/PeN] neurons evoked a slow EPSP in GnRH neurons in both parahorizontal (8.0 ± 2.9 mV, n = 4) (*Figure 11C–E*) and coronal slices

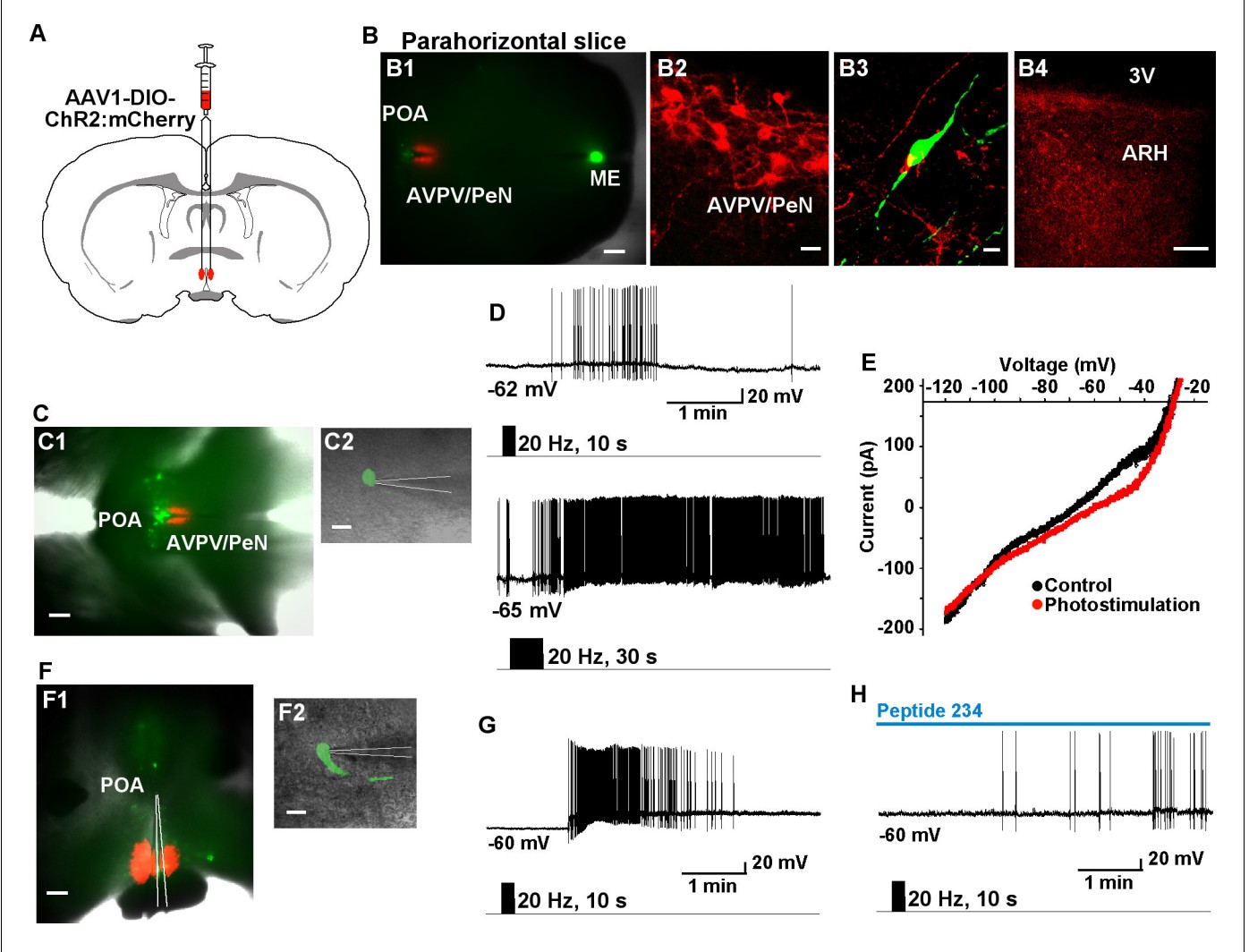

**Figure 11.** High-frequency photoactivation of Kiss1$^{AVPV/PeN}$ neurons excites GnRH neurons via kisspeptin release. (**A**) Schematic drawing of coronal section showing bilateral viral injections in the AVPV with AAV-DIO-ChR2:mCherry. (**B**) Parahorizontal brain slice of mouse bilaterally injected with a ChR2-mCherry-expressing viral vector into the AVPV/PeN of Kiss1-Cre::GnRH-EGFP female mouse. Injection sites into AVPV/PeN are illustrated in low power (red, **B1**) and high power (**B2**). Kiss1$^{AVPV/PeN}$ neurons send projections rostrally to GnRH neurons (**B3**) and also caudally to the ARH (**B4**). Note that following AVPV/PeN injections, only mCherry fibers but no labeled cells were detected in the ARH. Scale bars, 200 μM (**B1**), 20 μm (**B2**), 10 μm (**B3**) and 50 μm (**B4**). (**C**), Low power (**C1**) and high power (**C2**) images of whole-cell recordings in an angled, parahorizontal slice from bilateral AVPV AAV-injected Kiss1-Cre::GnRH-GFP mouse. (**D–E**) High-frequency photoactivation of AVPV/PeN with durations of 10 s (**upper**) and 30 s (**lower**) evoked a slow EPSP (mean slow EPSP response following 30 s duration stimulation: 8.0 ± 2.9 mV, n = 4) and increased firing (**D**); typical I-V relationship for a kisspeptin-evoked response in GnRH neurons in the same cell (**E**). Scale bars, 200 μm (**C1**) and 20 μm (**C2**). (**F**), Coronal slice showing ChR2-mCherry expression in Kiss1$^{AVPV/PeN}$ neurons following bilateral viral injections in the AVPV/PeN area in a Kiss1-Cre::GnRH-EGFP mouse (**F1**). Scattered GnRH-EGFP neurons can be seen dorsal and lateral to the injection site, one of which is being recorded from (**F1**, **low power**; **F2**, **high power**). Scale bars, 200 μm (**F1**) and 20 μm (**F2**). (**G**) High-frequency photostimulation of Kiss1$^{AVPV/PeN}$ neurons causes a slow EPSP in a GnRH neuron that induces a high-frequency discharge. (**H**), High-frequency-induced depolarization/excitation of GnRH neurons is blocked (occluded) by the partial agonist peptide 234 (100 nM). ARH, hypothalamic arcuate nucleus; AVPV/PeN, anteroventral periventricular and periventricular preoptic nuclei; ME, median eminence; POA, preoptic area.

(15.6 ± 4.8 mV, n = 4) (*Figure 11F–H*). The slow EPSP was mediated by the kisspeptin release from Kiss1$^{AVPV/PeN}$neurons based on the double rectifying, dual reversal potential I/V (*Figure 11E*) (*Zhang et al., 2008*) and occlusion of the response by the partial agonist peptide 234 (*Liu et al., 2011*), which blocked the evoked kisspeptin response by 85% (from 15.6 ± 4.8 mV to 2.4 ± 0.9 mV, n = 4) in coronal slices (*Figure 11H*). Finally, we harvested Kiss1$^{AVPV/PeN}$ neurons and measured

*Slc17a6* (vGluT2) mRNA expression using scRT-PCR. Based on the analysis of 71 Kiss1 neurons from 3 females (23–24 cells/ animal), only 2 cells (2.9% ) were found to express *Slc17a6* mRNA, an indication that glutamate is not packaged into vesicles and released by Kiss1[AVPV/PeN] neurons.

## Discussion

We have shown that high-frequency photostimulation of Kiss1[ARH] neurons expressing ChR2 evokes a slow excitatory postsynaptic potential (autoexcitation) in these Kiss1 neurons. The slow EPSP was abolished with Na$^+$ channel blocker TTX treatment but rescued by the addition of K$^+$ channel blockers, which is physiological evidence for Kiss1[ARH] synaptic contacts on Kiss1[ARH] neurons. Furthermore, the slow EPSP was attenuated by the Tacr3 antagonist SB222,200 and occluded by the Tacr3 agonist senktide pretreatment, demonstrating that the slow EPSP was mediated by the Gq-coupled Tacr3. In addition, the κ-opioid receptor agonist U69,593 markedly attenuated the slow EPSP, whereas the selective κ-antagonist nor-BNI enhanced the ChR2-evoked slow EPSP, indicative of κ-opioid receptor-mediated presynaptic inhibition of NKB release. As demonstrated with dual whole-cell recordings, there was a simultaneous activation and recruitment of Kiss1[ARH] neurons bilaterally to provide a synchronous drive to GnRH neurons. Finally, in horizontal but not the coronal hypothalamic slices, the Tacr3 agonist senktide indirectly excited GnRH neurons, and high-frequency photoactivation of Kiss1[ARH] neurons depolarized/excited GnRH neurons via a multisynaptic pathway that involved glutamatergic excitation of Kiss1[AVPV/PeN] neurons, which excited GnRH neurons via kisspeptin release.

High-frequency stimulation is required to evoke peptide release as originally demonstrated in the frog ganglion by *Jan et al. (1979)* (see *Arrigoni and Saper (2014)* for review). Indeed, high-frequency photostimulation of ChR2-expressing Kiss1[ARH] neurons elicited postsynaptic responses in neighboring Kiss1 neurons that entrained to stimulation frequencies up to 20 Hz. We observed a slowly developing depolarization over tens of seconds that was unaffected by ionotropic glutamate receptor blockade, but was antagonized by a Tacr3 antagonist. *Schöne et al. (2014)* evoked a slow EPSC in tuberomammillary histamine neurons following high-frequency (≥20 Hz) photostimulation of ChR2-expressing orexin neurons that was abrogated by a selective orexin receptor two antagonist. Therefore, in both kisspeptin and histamine neurons there is glutamate-dependent (low-frequency) and peptide-dependent (high-frequency) release that transmit distinct postsynaptic excitatory effects; but in the case of Kiss1[ARH] neurons, it is a recurrent auto-excitation that allows recruitment of neighboring Kiss1[ARH] neurons for synchronizing firing.

Also unique to Kiss1[ARH] neurons is that high-frequency stimulation co-released dynorphin, which modulated the NKB-Tacr3 response. Dynorphin is co-localized in the vast majority of Kiss1[ARH] neurons (*Navarro et al., 2009*; *Lehman et al., 2010a*), and loose-cell attached recordings have shown that dynorphin and κ-opioid receptor agonist U69,593 inhibit the activity of spontaneously firing Kiss1[ARH] neurons, although it has not been established whether this was a postsynaptic or presynaptic action on excitatory input to Kiss1[ARH] neurons (*de Croft et al., 2013*; *Ruka et al., 2013*). In our studies, dynorphin appeared to be co-released with high-frequency stimulation based on the fact that the slow EPSP evoked by high-frequency photostimulation was augmented with κ-opioid receptor antagonism (nor-BNI treatment), and abrogated by the κ-agonist U69,593. Moreover, although the κ-opioid receptor agonist U69,593 directly inhibits Kiss1[AVPV/PeN] neurons via activating G protein-coupled inwardly rectifying K$^+$ channels (*Zhang et al., 2013*), we did not see a similar direct response in Kiss1[ARH] neurons. However, the κ-agonist U69,593 did reduce the frequency of miniature EPSCs in Kiss1[ARH] neurons. Therefore, co-released dynorphin most likely acts presynaptically to modulate NKB release. This is analogous to the co-release of dynorphin from vasopressin neurons (*Iremonger and Bains, 2009*; *Iremonger et al., 2011*) in which it is co-packaged in the same dense core vesicles (*Shuster et al., 2000*). However, kisspeptin, NKB and dynorphin are packaged in separate vesicles as documented by electron microscopy (*Murakawa et al., 2015*) so there must be a coordinated co-release of the segregated peptide neurotransmitters (*Vaaga et al., 2014*) from Kiss1[ARH] neurons. Once released, dynorphin can inhibit synaptic vesicle mobilization/release via inhibiting presynaptic calcium channels (*Rusin et al., 1997*), activating K$^+$ channels (*Simmons and Chavkin, 1996*; *Vaughan and Christie, 1997*) or directly inhibiting the presynaptic release machinery (*Iremonger and Bains, 2009*). Presently, we do not know which one of the three mechanisms mediates the κ-opioid receptor-mediated actions on the presynaptic Kiss1[ARH] terminals, but it is clear that co-released dynorphin feeds back to modulate NKB release.

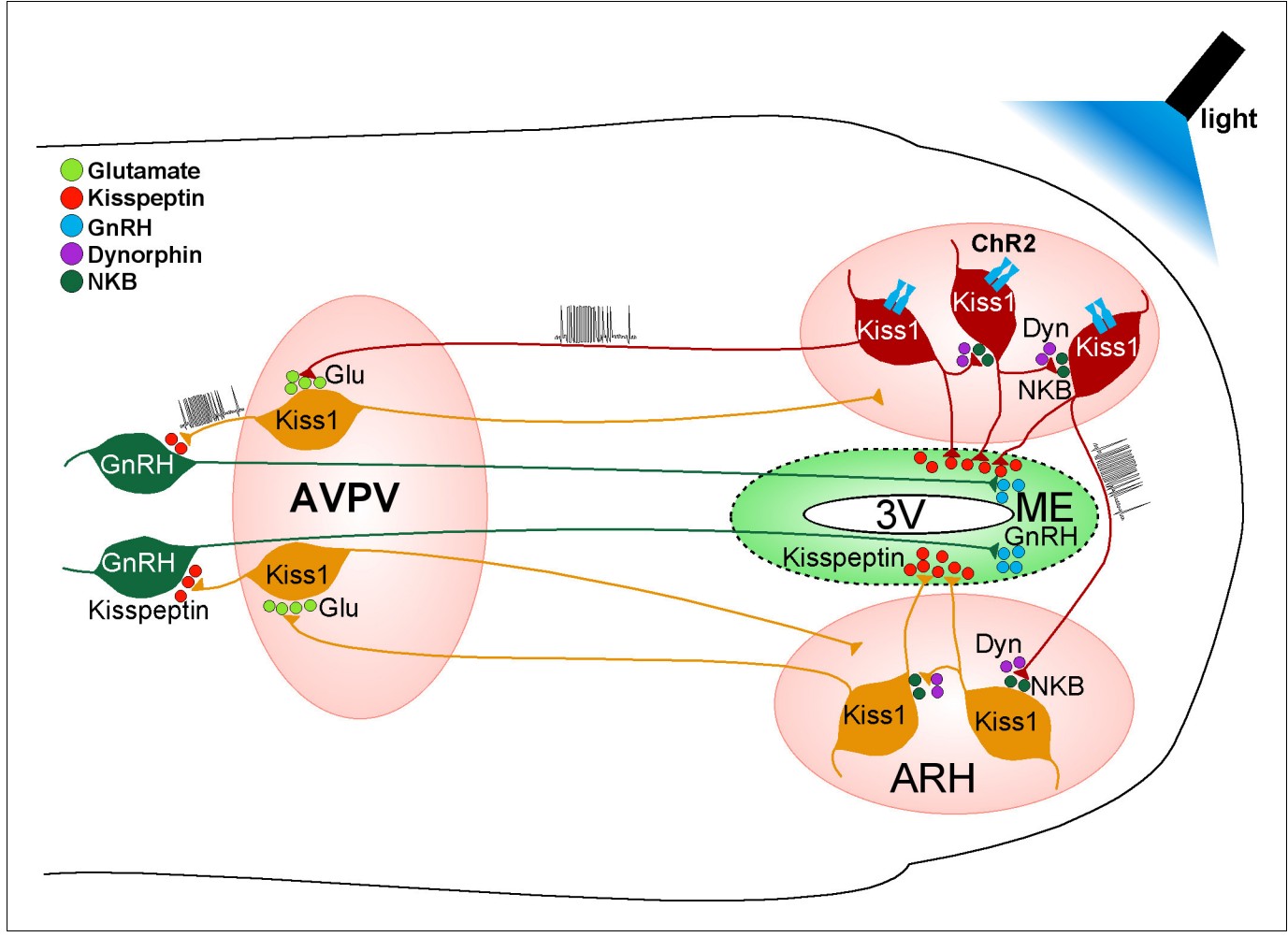

**Figure 12.** Proposed model by which activation of Kiss1 neurons governs GnRH neuronal excitability. High-frequency photostimulation of Kiss1 neurons in the ARH releases NKB that depolarizes and recruits other Kiss1$^{ARH}$ neurons. Dynorphin is co-released and acts presynaptically to modulate (inhibit) the release of NKB. Together the two peptides govern the synchronous activity of Kiss1$^{ARH}$ neurons and promote kisspeptin release that stimulates GnRH release in the median eminence (ME). Kiss1$^{ARH}$ neurons also communicate with the Kiss1$^{AVPV/PeN}$ neurons via the fast neurotransmitter glutamate, which stimulates burst-firing of Kiss1$^{AVPV/PeN}$ neurons. Activation of these rostral Kiss1 neurons releases kisspeptin to robustly excite GnRH neurons via activation of the GPR54 signaling cascade, thereby stimulating the release of GnRH at the time of the preovulatory surge. Kisspeptin, GPR54, NKB, Tacr3 and GnRH are all required for normal fertility.

It is known that nerve fibers expressing kisspeptin or co-expressing kisspeptin and NKB are localized in the internal zone of the median eminence in mice and rats (*Clarkson et al., 2009*; *True et al., 2010*), and a recent report has shown that simultaneous activation of Kiss1$^{ARH}$ neurons with optogenetic stimulation generates LH release in vivo, which is thought to be via kisspeptin-mediated depolarization of GnRH nerve terminals (*Han et al., 2015*). Moreover, morphological tract-tracing studies have shown that ARH NKB neurons, some of which are presumed to be Kiss1 neurons, are bilaterally interconnected, and NKB axons cross in the internal zone of the median eminence and project to the contralateral ARH (*Krajewski et al., 2010*). However, a direct synaptic connection between Kiss1$^{ARH}$ neurons has not been demonstrated. Herein, we show that ChR2:mCherry-labeled fibers cross over to the opposite side, and both a fast EPSC and a slow synaptic response (slow EPSP) were evoked with low- and high-frequency photostimulation, respectively, in the contralateral Kiss1$^{ARH}$ neurons. In fact, dual patch recording revealed simultaneous activation of Kiss1$^{ARH}$ neurons. Importantly, the contralateral Kiss1$^{ARH}$ neurons did not exhibit a direct ChR2 response thus confirming the confocal analysis that only ChR2 expressing fibers (axons) and control YFP expressing cell bodies were detected on the

contralateral side of the ARH. Moreover, the slow EPSP was blocked by Na$^+$ channel blocker TTX and rescued with the addition of K$^+$ blockers, which allowed robust depolarization of nerve terminals with photoactivation (*Cousin and Robinson, 2000*; *Petreanu et al., 2009*), and in our studies release of NKB. Therefore, it appears that Kiss1$^{ARH}$ neurons are contacted by other Kiss1$^{ARH}$ nerve terminals, and the activation of the Kiss1$^{ARH}$ network is dependent on action potential-driven synaptic release of NKB. Moreover, Kiss1$^{ARH}$ fibers crossing within the median eminence allow for synaptic communication between the arcuate nuclei for recruiting bilaterally additional Kiss1 neurons to generate a robust excitatory drive to GnRH neurons.

We used a horizontal slice preparation from Kiss1-Cre-ChR2::GnRH-EGFP mice, which contains both the ARH and AVPV/PeN Kiss1 neurons and preoptic GnRH neurons, to determine if an NKB agonist could excite GnRH neurons via a multi-synaptic pathway. Given that essentially all Kiss1$^{ARH}$ neurons, but not Kiss1$^{AVPV/PeN}$ or GnRH neurons, express Tacr3 mRNA (*Navarro et al., 2011b*; *Navarro et al., 2015*) and Kiss1$^{AVPV/PeN}$ neurons do not respond directly to kisspeptin (*Ducret et al., 2010*), we applied senktide to activate Kiss1$^{ARH}$ neurons and record changes in GnRH cell activity. Senktide induced robust firing in GnRH neurons in horizontal slices that included the Kiss1$^{ARH}$ neurons, but not in coronal slices excluding these neurons. These physiological findings support previous neuroanatomical tract tracing/immunocytochemical studies in mice showing that Kiss1$^{ARH}$ neurons project to AVPV/PeN (Kiss1) neurons but not to GnRH somata in the preoptic area (*Yip et al., 2015*). In addition, photoactivation of Kiss1$^{ARH}$ neurons excited Kiss1$^{AVPV/PeN}$ neurons via glutamate, which we have shown generates burst firing in these neurons (*Zhang et al., 2013*), and transsynaptically depolarized and excited GnRH neurons via kisspeptin activation of GPR54 and the opening of TRPC channels in a horizontal slice preparation that preserved all of the interconnections. These results are summarized in *Figure 12*. Therefore, our findings provide the first evidence that Kiss1$^{ARH}$ neurons can synchronize via NKB/Dyn auto-excitation/inhibition to provide an excitatory drive (via Kiss1$^{AVPV/PeN}$ neurons) to GnRH neurons for stimulating GnRH neuronal activity and subsequent GnRH release. This does not preclude the fact that Kiss1$^{ARH}$ neurons also project to the median eminence (*True et al., 2010*) and may stimulate pulsatile release of GnRH in the ovariectomized state (*Yip et al., 2015*). Hence, Kiss1$^{ARH}$ neurons probably play a dual role of driving pulsatile (episodic) secretion of GnRH through kisspeptin release into the external zone of the median eminence, but also excite Kiss1$^{AVPV/PeN}$ neurons via glutamate release to help generate the preovulatory surge of GnRH through a robust di-synaptic pathway (*Figure 12*). What is intriguing is the possibility that the differential release of amino acid and peptide neurotransmitters (segregated co-transmission) (*Vaaga et al., 2014*) occurs at these different release sites (ARH, median eminence, and AVPV/PeN) to coordinate reproductive function.

## Materials and methods

### Animals

Primarily female mice were used in this study. In a few instances male mice were also used for pharmacological analysis after it was determined that male and female neurons responded similarly. All animal procedures were conducted at Oregon Health and Science University (OHSU) according to the National Institutes of Health Guide for the Care and Use of Laboratory Animals and with approval from the OHSU Animal Care and Use Committee.

Kiss1-CreGFP mice (RRID:IMSR_JAX:017701) (*Gottsch et al., 2011*) were housed under constant temperature (21–23°C) and 12-h light, 12-h dark cycle schedule (lights on at 0600 and lights off at 1800 h), with free access to food (Lab Diets 5L0D) and water. Kiss1-CreGFP mice were used for viral injection to express ChR2 in ARH or AVPV/PeN Kiss1 neurons or they were crossed with heterozygous Ai32 mice (RRID:IMSR_JAX:024109, C57BL/6 background) purchased from The Jackson Laboratory. These Ai32 mice carry the *ChR2 (H134R)–EYFP* gene in their *Gt(ROSA)26Sor* locus (*Madisen et al., 2012*). The gene is separated from its CAG promoter by a *loxP*-flanked transcriptional STOP cassette, allowing its expression in a Cre-dependent manner. In the Kiss1-CreGFP::Ai32 (R26-flox-stop-ChR2 (H134R)-EYFP) cross, sometimes the Kiss1-CreGFP gene was expressed early in development resulting in expression of ChR2 (H134R)-EYFP gene ectopically in many cells (*Gottsch et al., 2011*). Those mice were not used for experiments. All other Kiss1-Ai32 crossed mice exhibited EYFP expression only in the expected locations (ARH and AVPV/PeN) as illustrated in

*Figure 2B,C* and were used for the experiments. In some studies, Kiss1-Cre:: GnRH-EGFP, AgRP-Cre::Ai32 and POMC-Cre::Ai32 female mice were used. These mice were produced by breeding Kiss1-CreGFP mice with GnRH-EGFP transgenic mice (*Suter et al., 2000*), and AgRP-Cre (RRID: IMSR_JAX:012899) or POMC-Cre mice (RRID:IMSR_JAX:010714) were bred with Ai32 mice, respectively.

## AAV delivery to Kiss1-CreGFP and Kiss1-Cre::GnRH-EGFP mice

Fourteen to twenty-one days prior to each experiment, Kiss1-CreGFP mice or Kiss1-Cre::GnRH-EGFP mice (>60 days old) received bilateral or unilateral ARH or AVPV injections of a Cre-dependent adeno-associated viral (AAV; serotype 1) vector encoding ChR2-mCherry (AAV1-Ef1α-DIO-ChR2: mCherry), ChR2-YFP (AAV1-Ef1α-DIO-ChR2:YFP) or YFP alone (AAV1-Ef1α-DIO-YFP). Using aseptic techniques, anesthetized female mice (1.5% isoflurane/$O_2$) received a medial skin incision to expose the surface of the skull. The glass pipette (Drummond Scientific #3-000-203-G/X; Broomall, PA) with a beveled tip (diameter = 45 µm) was filled with mineral oil, loaded with an aliquot of AAV using a Nanoject II (Drummond Scientific). ARH injection coordinates were anteroposterior (AP): −1.20 mm, mediolateral (ML): ± 0.30 mm, dorsoventral (DL): −5.80 mm (surface of brain z = 0.0 mm); AVPV injection coordinates were AP: 0.53 mm, ML: ± 0.30 mm, DL: −4.70 and −5.10 mm (surface of brain z = 0.0 mm); 500 nl of the AAV (2.0 x $10^{12}$ particles/ml) was injected (100 nl/min) into each position, left in place for 10 min post-injection, then the pipette was slowly removed from the brain. The skin incision was closed using skin adhesive, and each mouse received analgesia (Rimadyl; 4 mg/kg) for two days post-operation.

It should be noted that the fluorescence intensity of YFP in Kiss1[ARH] neurons following viral injections or as a result of Kiss1-CreGFP::Ai32 cross masked the weaker GFP signal in Kiss1-CreGFP neurons, i.e. no weak GFP cells were observed but only intensely labeled YFP cells. Therefore, in initial experiments these ARH YFP cells were dispersed and harvested, or harvested in the slice following whole cell recording and subsequently subjected to single cell (sc)RT-PCR (see below). In all instances, the ARH YFP cells were found to express Kiss1 mRNA (see *Figure 2*).

## Experimental design

Females were used with gonads intact or were subjected to ovariectomy (OVX). It is well known that OVX animals exhibit increased GnRH and LH pulsatile secretion that coincide with increased expression of neuropeptides (including kisspeptin, neurokinin B and dynorphin) in ARH Kiss1 neurons (*Caraty et al., 1989*; *Smith et al., 2005*; *Dellovade and Merchenthaler, 2004*; *Gottsch et al., 2009*). Since we were interested in determining the cellular mechanism by which ARH Kiss1 neurons synchronize to induce pulsatile GnRH and LH secretion, as well as determining the stimulus needed for peptide release, we primarily used OVX females for these experiments. However, we also used intact females at different stages of the estrous cycle in order to compare the magnitude of the high-frequency response from these animals to the response in OVX females (see *Figure 2*). For studies involving AVPV/PeN Kiss1 neurons and GnRH neurons, we used OVX females treated with E2 as a model for the proestrous stage of the cycle (*Zhang et al., 2013*, *2015*; *Rønnekleiv et al., 2015*).

## Gonadectomy

Seven days prior to each experiment, ovariectomies were performed on female mice under inhalant isofluorane anesthesia. Carprofen (Rimadyl; Pfizer) was given immediately after a surgery at a dose of 4 mg/kg for relief of postoperative pain. Two to three days prior to electrophysiological experiments, OVX females were given oil-vehicle injections or low dose followed by a high dose one day later of 17β-estradiol (s.c.), which produces an LH surge on the experimental day (*Bosch et al., 2013*).

## Immunocytochemistry

Female mice were prepared for immunocytochemistry as described previously (*Roepke et al., 2011*). Briefly, two mm coronal hypothalamic blocks were fixed by immersion in 4% paraformaldehyde, cryoprotected in 20% sucrose solution, frozen at −55°C, sectioned coronally on a cryostat at 20 µm, and thaw-mounted on Superfrost Plus slides (Thermo Fisher Scientific). Sections were rinsed

in PB (0.1 M phosphate buffer, pH 7.4; all rinses were in PB for at least 30 min), and then incubated for 24 hr at 4°C in rabbit polyclonal antiserum against mCherry (1:5000–1:10,000; ab167453 Abcam Inc, Cambridge MA). Some sections were also reacted with goat polyclonal antiserum against GFP conjugated to FITC (RRID:AB_305635) (1:1000; ab6662 Abcam). After rinsing, sections stained for mCherry were incubated in donkey-antirabbit IgG antibody conjugated to Cy3 (1:300; Jackson Immunoresearch). Following a final rinse, slides were coverslipped using vectashield (Vector Laboratories).

## Imaging

Immunocytochemical-stained sections and slices used for recording were acquired and analyzed using confocal microscopy. Photomicrographs were acquired by an Olympus BX51W1 upright microscope equipped with a Rolera XR Fast 1394 camera. Confocal photomicrographs were acquired using a Zeiss LSM 510 and a Zeiss LSM 780 confocal microscopes with Zen software (RRID:SCR_013672).

## Electrophysiology

Coronal brain slices (250 μm) containing the ARH or AVPV/PeN from gonadectomized or intact mice were prepared as previously described (*Qiu et al., 2003*). Also in some experiments, angled para-horizontal brain slices were cut as previously described (*Liu et al., 2011*), and the vibratome blade was positioned to just touch the optic chiasm and a single 400 μm-thick horizontal slice prepared. Whole-cell, patch recordings were performed in voltage clamp and current clamp using an Olympus BX51W1 upright microscope equipped with video-enhanced, infrared-differential interference contrast (IR-DIC) and an Exfo X-Cite 120 Series fluorescence light source. Electrodes were fabricated from borosilicate glass (1.5 mm outer diameter; World Precision Instruments, Sarasota, FL) and filled with a normal internal solution (in mM): 128 potassium gluconate, 10 NaCl, 1 $MgCl_2$, 11 EGTA, 10 HEPES, 2 ATP, and 0.25 GTP (pH was adjusted to 7.3–7.4 with 1N KOH, 290–300 mOsm). Pipette resistances ranged from 3–5 MΩ. In whole cell configuration, access resistance was less than 20 MΩ; access resistance was 80% compensated. For optogenetic stimulation, a light-induced response was evoked using a light-emitting diode (LED) 470 nm blue light source controlled by a variable 2A driver (ThorLabs, Newton, NJ) with the light path delivered directly through an Olympus 40 × water-immersion lens. High fidelity response to light (470 nm) stimulation of Kiss1$^{ARH}$ ChR2-expressing neurons was observed, and both evoked inward currents (in voltage clamp, $V_{hold}$ = −60 mV) or depolarization (in current clamp) were measured (*Figures 1* and *2*). Electrophysiological signals were amplified with an Axopatch 200A and digitized with Digidata 1322A (Molecular Devices, Foster City, CA), and the data were analyzed using p-Clamp software (RRID:SCR_011323, version 9.2, Molecular Devices). A MultiClamp 700B amplifier (Molecular Devices, Foster City, CA) was used for the dual patch recording, and the signals were similarly digitized and analyzed. The amplitude of the slow EPSP was measured after low pass filtering in order to eliminate the barrage of action potentials riding on the depolarization (*Figure 2E*). The liquid junction potential was corrected for all data analysis.

## Single cell RT-PCR

After electrophysiological recording, the cytosol of recorded arcuate YFP-expressing cells from Kiss1-Cre::Ai32 mice was harvested and used for post-hoc identification of *Kiss1* mRNA. Single-cell harvesting and RT-PCR was conducted as described (*Bosch et al., 2013*; *Zhang et al., 2013*). Briefly, the recorded cells were harvested with gentle suction to the recording pipette and expelled into a siliconized 0.65 ml microcentrifuge tube containing a solution of 1X Invitrogen Superscript III Buffer, 15U of RNasin (Promega), 10 mM of dithiothreitol (DTT) and diethylpyrocarbonate (DEPC)-treated water in a total of 5 μl for a single cell. cDNA synthesis was performed on single cells and stored at −20°C. Controls included harvested cells processed without reverse transcriptase (−RT) and hypothalamic tissue RNA reacted with and without reverse transcriptase. Primers were designed using Clone Manager software (Sci Ed Software) to cross at least one intron-exon boundary and optimized as previously described (*Bosch et al., 2013*). The primers for *Kiss1* were as described previously (*Zhang et al., 2013*). In addition, Kiss1 neurons in the preoptic area (POA) were dispersed and harvested as described (*Zhang et al., 2013*) and the mRNA expression of the vesicular glutamate

transporter 2 (vGluT2) was analyzed. The *Slc17a6* (vGluT2) primers were as follows: 136 bp product, accession number NM_080853, forward primer 1677–1694 nt, reverse primer 1795–1812 nt. Single-cell RT-PCR was performed on 3 µl of cDNA in a 30 µl reaction volume and amplified 40–50 cycles using a C1000 Thermal Cycler (Bio-Rad). The PCR products were visualized with ethidium bromide on a 2% agarose gel. The scRT-PCR products for *Kiss1* and *Slc17a6* (vGluT2) have been confirmed by sequencing.

## Drugs

All drugs were purchased from Tocris Bioscience (Minneapolis, MN) unless otherwise specified. DL-2-amino-5-phosphonopentanoic acid sodium salt (AP5) (50 mM), 6-cyano-7-nitroquinoxaline-2, 3-dione disodium (CNQX) (10 mM), 4-Aminopyridine (4-AP) (500 mM) and Nor-Binaltorphimine dihydrochloride (1 mM) and were dissolved in $H_2O$. Tetrodotoxin (TTX) was purchased from Alomone Labs (Jerusalem, Israel) (1 mM) and dissolved in $H_2O$. Tetraethylammonium chloride (TEA, 7.5 mM) and U69,593 (20 mM) were purchased from Sigma-Aldrich (St. Louis, MO) and dissolved in aCSF or ethanol, respectively. Tacr1 antagonist SDZ-NKT-343 (20 mM), Tacr2 antagonist GR94,800 (2 mM), Tacr3 antagonist SB222,200 (50 mM) and Tacr3 agonist senktide (1 mM) and $GABA_A$ receptor antagonist Bicuculline methiodide (50 mM) were prepared in DMSO. Guanosine 5′-[β-thio] diphosphate trilithium salt (GDP-β-S, 2 mM) from Calbiochem (La Jolla, CA) and QX 314 chloride (0.5 mM) were dissolved in pipette solution. Kisspeptin-10 [Mouse Kiss-1(110–119)-NH2; Kp-10, from Phoenix Pharmaceuticals (Belmont, CA)] (100 µM) and peptide 234 (2 mM) (gift from Dr. Robert Millar, U. Pretoria, South Africa) were dissolved in $H_2O$. Aliquots of the stock solutions were stored as appropriate until needed.

## Data analysis

Comparisons between different treatments were performed using a one-way ANOVA analysis with the Newman-Keuls's post hoc test. Differences were considered statistically significant if $p < 0.05$. All data are expressed as mean ± SEM.

## Acknowledgements

We thank Ms. Martha A Bosch and Ms. Uyen-Vy Navarro for excellent technical support. We thank Drs. Edward J Wagner (Western University of Health Sciences) and Wei Fan (Oregon Health & Science University) for their comments on an earlier draft of the manuscript. The POMC-Cre mice were obtained from the Jackson Laboratory, the AgRP-Cre mice were obtained from Dr. Joel Elmquist (UT Southwestern) and the GnRH-GFP mice from Dr. Suzanne Moenter (U Michigan). We thank Dr. Robert P Millar (U. Pretoria, South Africa) for the generous gift of peptide 234. This research was funded by National Institute of Health (NIH) grants R01-NS38809 (MJK), R01-NS43330 (OKR), R01-DK68098 (MJK and OKR) and F32 DK104366 (CCN). Confocal microscopy was supported by the P30 NS061800 grant and pilot funding for CCN from the University Shared Resources program.

## Additional information

### Competing interests

RDP: Reviewing editor, *eLife*. The other authors declare that no competing interests exist.

### Funding

| Funder | Grant reference number | Author |
| --- | --- | --- |
| National Institutes of Health | R01-NS38809 | Martin J Kelly |
| National Institutes of Health | R01-NS43330 | Oline K Rønnekleiv |
| National Institutes of Health | R01-DK68098 | Martin J Kelly<br>Oline K Rønnekleiv |
| National Institutes of Health | F32 DK104366 | Casey C Nestor |

The funders had no role in study design, data collection and interpretation, or the decision to submit the work for publication.

## Author contributions

JQ, Designed the study, Performed and analyzed electrophysiological data, Wrote the manuscript with input from all contributing authors; CCN, Designed the study, Did the viral injections and performed the molecular biology experiments, Performed the histological and microscopy analysis; CZ, Performed and analyzed electrophysiological data; SLP, RDP, Provided reagents; MJK, Designed the study, Provided laboratory space and resources to conduct the experiments, Wrote the manuscript with input from all contributing authors; OKR, Designed the study, Performed the histological and microscopy analysis, Provided laboratory space and resources to conduct the experiments, Wrote the manuscript with input from all contributing authors

## Author ORCIDs

Jian Qiu, http://orcid.org/0000-0002-4988-8587

## Ethics

Animal experimentation: This study was performed in strict accordance with the recommendations from the National Institutes of Health Guide for the Care and Use of Laboratory Animals. All animal procedures were conducted according to the approved institutional animal care and use committee (IACUC) protocols (#IS00003005; #IP00000382) at Oregon Health and Science University. All surgeries were performed using aseptic techniques under isoflurane anesthesia, and every effort was made to minimize suffering.

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
