## [Decision Letter]

Thank you for submitting your article "High frequency stimulation-induced peptide release synchronizes arcuate kisspeptin neurons and excites GnRH neurons" for consideration by *eLife*. Your article has been favorably evaluated by a Senior Editor and three reviewers, one of whom is a member of our Board of Reviewing Editors.

The reviewers have discussed the reviews with one another and the Reviewing Editor has drafted this decision to help you prepare a revised submission.

Since the report by Belchetz et al. (1978), it has been known that the delivery of GnRH1 to the pituitary must be pulsatile to be effective. In the intervening 38 years, there has been no clear answer about how this is generated. The discovery of kisspeptin peptide (Kiss1) led to the hypothesis that these neurons might be a clue to the pulsatile production of GnRH1. The authors provide a clear and balanced view of the progress in understanding (or lack thereof) and used a collection of techniques to suggest a putative microcircuitry underlying the Kiss1/GnRH regulation. In rodents, Kiss 1 neurons are interconnected in the hypothalamic arcuate nucleus (ARH) and project GnRH1 terminals in the median eminence. The essential finding is that high frequency photoactivation of Kiss1 neurons in the ARH induced a NKB-mediated slow excitatory postsynaptic potential, causing synchronization of Kiss1 neurons that in turn excites GnRH neurons. The results are interesting and significant because the many optogenetic experiments carried out in peptidergic neurons have not generally revealed peptide release.

This paper also provides evidence of a possible peptidergic circuit linking the arc KINDY neurons to those in the AVPV. One open question is whether there is evidence here for axonal release as opposed possible somatodendritic release activating autoreceptors to cause release of other additional retrograde messengers. Overall, the reviewers agree that this is a technical tour de force, providing a significant step forwards in understanding the very important reproductive circuit.

A number of questions need to be addressed in the revision process. Please consider the following suggestions and perform the requested experiments or rebut them and clarify as needed:

1) Figure 3 figure legend and text (subsection “The slow EPSP is dependent on the direct synaptic input from Kiss1 neurons”). It is possible that the addition of TEA enables sufficient calcium to enter the cell to evoke local somatodendritic release from the cell under study that activates autoreceptors; it may not be necessary to have release from axons of neighboring cells. This is also true for Figure 7 where K + blockers rescued the EPSP.

2).In Figure 4, you show that the EPSP is not evoked by electrically induced by action potentials in the soma. However, it is possible that they are not invading the terminals, in contrast to light evoked activation that will depolarize the entire membrane. What would happen if this were done in the presence of 4-AP/TEA? It also raises the obvious question – cells normally fire action potentials that are not evoked optogenetically. Are your responses an optogenetic artefact? I mean not that the findings are artefactual, but that they would not occur in the intact animal?

3).Figure 6 – raw data do not graphically support your contention that norBNI potentiates the slow EPSP and is not representative of the statement stipulating "that blockade of κ-opioid receptor by the selective antagonist nor-BNI (1 μM) potentiated the slow EPSP". One cannot see any potentiation (increase in amplitude) of the slow EPSP in this figure. Figure 6 – are these two different cells? It is interesting that you claim (not surprisingly) that U69,593 reduces mEPSP frequency, yet the lower of these figures appears to show unaltered minis. Did U69,593 affect the postsynaptic properties of the cell under study (aside from blocking the slow EPSP)? What happened to input resistance? Differences in the frequency of the mEPSCs are not visible in Figure 6. Could the authors find a better illustration and expand the time scale of the mEPSCs recordings?

4) Figure 7, you suggest that fibres from the contralateral side cross the midline and activate contralateral KINDY neurons. However, how do you know that the unilateral injection of CHR2 did not also express in fibres crossing over from cell bodies on the other side, thus having CHR2 expressed bilaterally? Thus you could have back propagating spikes activating the contralateral neurons and thus release local tachykines to cause the slow EPSP by depolarizing axons of neurons from the other side that express ChR2 and back-propagating spikes cause release of NKB from the dendrites of Kiss-1 cells. This must at least be acknowledged (or alternately, a clear statement made here that there was no mcherry detected in any of the YFP expressing neurons – I am aware that you partially address this in the Methods but it is critical that it be stated here). This is problematic as you state that the fluorescence from the YFP was very much brighter than the mCherry, making it possible that you overlook the possible mCherry expression.

5) The paragraph of the results dealing with the role of Tacr3 and the slow EPSP (subsection “The slow EPSP is mediated by the G protein-coupled receptor Tacr3”) needs some work: what is the rationale? Why do both the antagonist and the agonist of Tacr3 block the slow EPSP? "Therefore" is misleading (has it been shown before?). The sentence – “To see if dual photostimulation induced desensitization, we perfused senktide repeatedly (2x), and the averaged ratio of the responses (0.65 ± 0.01, n = 4) was similar to that obtained by photostimulation, suggesting that desensitization was caused by NKB-mediated downregulation of Tacr3 (Figure 5)” – is difficult to understand; did the authors mean that two successive applications of senktide mimicked the desensitization induced by dual photostimulation? The effect of dual photostimulation is not illustrated: it should be shown in Figure 5. The authors state that their results suggest that desensitization was caused by NKB-mediated downregulation of Tacr3 expression. Why not its internalization, its desensitization, etc.? Do the authors have data supporting this statement? What does GDP-β-S stands for; what is its action?

6) Are the effects of the photostimulation of the contralateral Kisspeptinergic neuronal fibers also blocked with the Tacr3 antagonist or the k-opioid receptor agonist in the ARH?

7) Do the authors have similar results when they stimulate contralateral cell bodies?

8) The authors state that photostimulation of contralateral ARH Kiss neurons induce synchronization of Kisspeptinergic neuronal populations in the ipsilateral ARH (subsection “Kiss1^ARH^ neurons directly communicate with contralateral Kiss1^ARH^ neurons” in the Results and in the fifth paragraph of the Discussion). Testing this hypothesis experimentally (ex: using dual patch-clamp recordings) would be very important for strengthening the story and proposed mechanisms.

9) Further pharmacological characterization would be required Figure 11 (in addition to the I/V curve) to actually show that evoked slow EPSPs in GnRH neurons are mediated by kisspeptin (ex: using kisspeptin antagonists) (subsection “Kiss1^ARH^ neurons excite AVPV/PeN Kiss1 neurons via glutamate”).

10) The authors convincingly demonstrate that ARH Kiss neurons provide an excitatory drive to GnRH neurons via AVPV kisspeptin neurons. However, the physiological significance of this exciting finding with respect to GnRH release is not clear. If the authors could show that by inhibiting ARH kisspeptin neurons on proestrus (or in OVX + E2 mice), it decreases surge levels of LH (surrogate measure of GnRH secretion), it would be a great addition to the present study.

11) Kiss-1 neurons in the AVPV are apparently driving activity in GnRH neurons through release of kisspeptin. As these neurons presumably also contain tachykines and dynorphin, could NK3 or opioid receptors also mediate part of the effect? Does single cell PCR of GNRH neurons reveal either opioid or NK3 receptors? It would be useful to apply antagonists to these peptides and ensure that the excitation persists.

---

## [Author Response]

*A number of questions need to be addressed in the revision process. Please consider the following suggestions and perform the requested experiments or rebut them and clarify as needed:*

1) Figure 3 figure legend and text (subsection “The slow EPSP is dependent on the direct synaptic input from Kiss1 neurons”). It is possible that the addition of TEA enables sufficient calcium to enter the cell to evoke local somatodendritic release from the cell under study that activates autoreceptors; it may not be necessary to have release from axons of neighboring cells. This is also true for Figure 7 where K + blockers rescued the EPSP.

The reviewer is correct that when recordings are from the ipsilateral ARC in which the ChR2 virus is injected there could be somatodendritic release of peptides. However, the contralateral ARC recordings in which no ChR2 virus was injected and no ChR2-labeled cell bodies or light-evoked currents were observed argues against this scenario since only Kiss1 axons cross over the median eminence to the contralateral ARC (Lehman et al. 2010; Rance et al. 2010). This was the rationale for the experiments described in the Results (subsection “Kiss1^ARH^ neurons directly communicate with contralateral Kiss1^ARH^ neurons”) and depicted in Figure 7.

2) In Figure 4, you show that the EPSP is not evoked by electrically induced by action potentials in the soma. However, it is possible that they are not invading the terminals, in contrast to light evoked activation that will depolarize the entire membrane. What would happen if this were done in the presence of 4-AP/TEA? It also raises the obvious question – cells normally fire action potentials that are not evoked optogenetically. Are your responses an optogenetic artefact? I mean not that the findings are artefactual, but that they would not occur in the intact animal?

We have now included in the Results (subsection “The slow EPSP is dependent on the direct synaptic input from Kiss1 neurons”) the fact that high frequency electrical stimulation of individual neurons in the presence of 4AP/TEA does not evoke a slow EPSP (Figure 4). We believe that the sEPSP is the result of activation of many neurons and underlies synchronization and pulse generation in these neurons, a phenomenon that was first demonstrated in multiunit recordings in intact animals (Knobil 1989).

*3) Figure 6 – raw data do not graphically support your contention that norBNI potentiates the slow EPSP and is not representative of the statement stipulating "that blockade of κ-opioid receptor by the selective antagonist nor-BNI (1 μM) potentiated the slow EPSP". One cannot see any potentiation (increase in amplitude) of the slow EPSP in this figure.*

We have now provided a more representative figure (Figure 6) that supports this statement as well as the summary data.

*Figure 6 – are these two different cells? It is interesting that you claim (not surprisingly) that U69,593 reduces mEPSP frequency, yet the lower of these figures appears to show unaltered minis.*

These recordings were from the same cell. We now provide a new Figure 6 that shows that U69,593 blocks the slow EPSP (B2 versus B1), but the slow EPSP can still be evoked pharmacologically by applying senktide.

*Did U69,593 affect the postsynaptic properties of the cell under study (aside from blocking the slow EPSP)? What happened to input resistance?*

U69,593 did not induce an outward current in Kiss1^ARH^ neurons and did not change the input resistance. We now state this in the Results (subsection “The slow EPSP is modulated by endogenous dynorphin”).

Differences in the frequency of the mEPSCs are not visible in Figure 6. Could the authors find a better illustration and expand the time scale of the mEPSCs recordings?

We have included another example with an expanded time scale (Figure 6) to illustrate the inhibition of mEPSCs by U69,593.

4) Figure 7, you suggest that fibres from the contralateral side cross the midline and activate contralateral KINDY neurons. However, how do you know that the unilateral injection of CHR2 did not also express in fibres crossing over from cell bodies on the other side, thus having CHR2 expressed bilaterally? Thus you could have back propagating spikes activating the contralateral neurons and thus release local tachykines to cause the slow EPSP by depolarizing axons of neurons from the other side that express ChR2 and back-propagating spikes cause release of NKB from the dendrites of Kiss-1 cells. This must at least be acknowledged (or alternately, a clear statement made here that there was no mcherry detected in any of the YFP expressing neurons – I am aware that you partially address this in the Methods but it is critical that it be stated here). This is problematic as you state that the fluorescence from the YFP was very much brighter than the mCherry, making it possible that you overlook the possible mCherry expression.

[We apologize for the misunderstanding, but we never stated that the YFP was much brighter than the mCherry. Our statement in the Methods was that YFP from the viral injections was much brighter than the GFP within Kiss1-GFP neurons and masked the endogenous GFP label. We used this as a rationale for harvesting YFP neurons and using scRT-PCR to determine that they expressed Kiss1 mRNA.]

We have addressed this concern in two ways (anatomically and electrophysiologically). First, we used an antibody to boost mCherry expression on the contralateral side of a unilaterally injected Kiss1-Cre mouse and only found intensified fiber staining and no staining of cell somas. We have provided a confocal image to illustrate this point (Figure 7). Second, in all of whole-cell recordings we did not see any ChR2-induced depolarizations (currents) in the contralateral ARC Kiss1 neurons after unilateral ARC injections. We have stated this in the Results (subsection “Kiss1^ARH^ neurons directly communicate with contralateral Kiss1^ARH^ neurons”).

5) The paragraph of the results dealing with the role of Tacr3 and the slow EPSP (subsection “The slow EPSP is mediated by the G protein-coupled receptor Tacr3”) needs some work: what is the rationale? Why do both the antagonist and the agonist of Tacr3 block the slow EPSP? "Therefore" is misleading (has it been shown before?).

We have clarified in the Results that Tacr3 agonist senktide depolarizes Kiss1^ARH^ neurons and pharmacologically prevents any effects of the photostimulated release of NKB (subsection “The slow EPSP is mediated by the G protein-coupled receptor Tacr3”).

*The sentence – “To see if dual photostimulation induced desensitization, we perfused senktide repeatedly (2x), and the averaged ratio of the responses (0.65 ± 0.01, n = 4) was similar to that obtained by photostimulation, suggesting that desensitization was caused by NKB-mediated downregulation of Tacr3 (Figure 5)” – is difficult to understand; did the authors mean that two successive applications of senktide mimicked the desensitization induced by dual photostimulation?*

Yes, two successive applications of senktide mimicked the desensitization. We have re-worded the sentence to make it more clear: “To see if dual photostimulation induced desensitization, we perfused senktide repeatedly (2x), and the averaged ratio of the responses (0.65 ± 0.01, n = 4) was similar to that obtained by photostimulation, suggesting that desensitization was caused by internalization and downregulation of Tacr3 following binding of NKB (Steinhoff et al., 2014) (Figure 5).”.

*The effect of dual photostimulation is not illustrated: it should be shown in Figure 5.*

We have shown the dual photostimulation in Figure 3. But we now have added another trace in Figure 5 to emphasize this point.

*The authors state that their results suggest that desensitization was caused by NKB-mediated downregulation of Tacr3 expression. Why not its internalization, its desensitization…etc? Do the authors have data supporting this statement?*

We were using “desensitization” in the general sense and certainly, it would include internalization. We are basing this statement on the general knowledge of how tachykinin receptors behave and have included a reference (subsection “The slow EPSP is mediated by the G protein-coupled receptor Tacr3”).

What does GDP-β-S stands for; what is its action?

GDP-β-S stands for Guanosine 5′-[β-thio]diphosphate trilithium salt. It is a GDP analog that inhibits G protein activation. We have added this to the Results (subsection “The slow EPSP is mediated by the G protein-coupled receptor Tacr3”).

6) Are the effects of the photostimulation of the contralateral Kisspeptinergic neuronal fibers also blocked with the Tacr3 antagonist or the k-opioid receptor agonist in the ARH?

Yes, and we now include a whole-cell recording of the Tacr3 antagonist blockade in the Results (subsection “Kiss1^ARH^ neurons directly communicate with contralateral Kiss1^ARH^ neurons”) and in Figure 7.

7) Do the authors have similar results when they stimulate contralateral cell bodies?

If we understand the reviewer’s question correctly, we block the whole-cell response (slow EPSP) in the contralateral ARC Kiss1 neurons with SB222,200 (Results, subsection “Kiss1^ARH^ neurons directly communicate with contralateral Kiss1^ARH^ neurons”).

8) The authors state that photostimulation of contralateral ARH Kiss neurons induce synchronization of Kisspeptinergic neuronal populations in the ipsilateral ARH (subsection “Kiss1^ARH^ neurons directly communicate with contralateral Kiss1^ARH^ neurons” in the Results and in the fifth paragraph of the Discussion). Testing this hypothesis experimentally (ex: using dual patch-clamp recordings) would be very important for strengthening the story and proposed mechanisms.

We now include examples of dual patch recording of arcuate Kiss1 neurons in which photostimulation induced depolarization (slow EPSPs) and increased firing simultaneously (new Figure 8).

9) Further pharmacological characterization would be required Figure 11 (in addition to the I/V curve) to actually show that evoked slow EPSPs in GnRH neurons are mediated by kisspeptin (ex: using kisspeptin antagonists) (subsection “Kiss1^ARH^ neurons excite AVPV/PeN Kiss1 neurons via glutamate”).

Unfortunately, there is currently no bona fide full antagonist to kisspeptin. At best, peptides 234, 271, 318 are partial agonists in vitro (Robert Millar, personal communication). However, by using peptide 234 to pharmacologically occlude the actions of kisspeptin, we now show that we can block the effects of high frequency photostimulation in the presence of the partial agonist peptide 234 (new Figure 12).

*10) The authors convincingly demonstrate that ARH Kiss neurons provide an excitatory drive to GnRH neurons via AVPV kisspeptin neurons. However, the physiological significance of this exciting finding with respect to GnRH release is not clear. If the authors could show that by inhibiting ARH kisspeptin neurons on proestrus (or in OVX + E2 mice), it decreases surge levels of LH (surrogate measure of GnRH secretion), it would be a great addition to the present study.*

This indeed would be an elegant experiment, but it is beyond the scope of this study. We have plans to explore both GnRH release in vitro and LH release in vivo in the future. However, this is no small task given that it involves developing a serial blood-sampling procedure in mice as well as sensitive assays for measuring GnRH release in vitro and LH in very small quantities of blood. At least the Herbison lab has shown that optogenetic stimulation of the arcuate kisspeptin neurons releases LH in males and ovariectomized females (Han et al., 2015).

*11) Kiss-1 neurons in the AVPV are apparently driving activity in GnRH neurons through release of kisspeptin. As these neurons presumably also contain tachykines and dynorphin, could NK3 or opioid receptors also mediate part of the effect?*

AVPV Kiss1 neurons co-express TH, but not NKB or substance P. They also express κ-opioid receptors, but to our knowledge co-expression of dynorphin has not been documented in AVPV Kiss1 neurons.

Does single cell PCR of GNRH neurons reveal either opioid or NK3 receptors?

Based on our sensitive scRT-PCR, very few GnRH neurons express NK3 receptors (5-10% versus 100% in arcuate Kiss1 neurons; Navarro et al., Endocrinology 2014). Also, we have demonstrated (Figure 9) that GnRH neurons in coronal slices, which are cut-off from the arcuate nucleus, do not respond to senktide (i.e. they do not express the NK3 receptor Tacr3).

*It would be useful to apply antagonists to these peptides and ensure that the excitation persists.*

As mentioned above, GnRH neurons do not respond to the NK3R agonist senktide (Figure 9) or the kappa-receptor agonist U69,593 (unpublished findings).